# Spatio-Temporal Approximation: A Training-Free SNN Conversion for Transformers

**Yizhou Jiang**[1][*]**, Kunlin Hu**[2][*]**, Tianren Zhang**[1]**, Haichuan Gao**[1]**, Yuqian Liu**[1]**,
Ying Fang**[3][†]**, Feng Chen**[1,4][†]

[1]Department of Automation, Tsinghua University, Beijing, China
[2]Tsinghua Shenzhen International Graduate School, Tsinghua University, Shenzhen, China
[3]College of Computer and Cyber Security, Fujian Normal University, Fuzhou, China
[4]LSBDPA Beijing Key Laboratory, Beijing, China
`{jiangyz20, hukl22, zhangtr22}@mails.tsinghua.edu.cn,`
`ghc2023@mail.tsinghua.edu.cn, liuyuqian21@mails.tsinghua.edu.cn,`
`fy20@fjnu.edu.cn, chenfeng@mail.tsinghua.edu.cn`

## Abstract

Spiking neural networks (SNNs) are energy-efficient and hold great potential for large-scale inference. Since training SNNs from scratch is costly and has limited performance, converting pretrained artificial neural networks (ANNs) to SNNs is an attractive approach that retains robust performance without additional training data and resources. However, while existing conversion methods work well on convolution networks, emerging Transformer models introduce unique mechanisms like self-attention and test-time normalization, leading to non-causal non-linear interactions unachievable by current SNNs. To address this, we approximate these operations in both temporal and spatial dimensions, thereby providing the first SNN conversion pipeline for Transformers. We propose *Universal Group Operators* to approximate non-linear operations spatially and a *Temporal-Corrective Self-Attention Layer* that approximates spike multiplications at inference through an estimation-correction approach. Our algorithm is implemented on a pretrained ViT-B/32 from CLIP, inheriting its zero-shot classification capabilities, while improving control over conversion losses. To our knowledge, this is the first direct training-free conversion of a pretrained Transformer to a purely event-driven SNN, promising for neuromorphic hardware deployment. Codes are available at https://github.com/ViviaHu/STA.

## 1 Introduction

The recent success of large Transformer models has increased the need for efficient inference. Spiking neural networks (SNNs), as the third generation of neural networks, use multi-step sparse spike accumulations instead of dense multiply-accumulations, providing significant advantages in energy and speed. This makes SNNs a prospective candidate to replace ANNs for large-scale deployment.

Due to the non-differentiability of spiking neurons, obtaining large-scale SNNs remains a challenge. Existing method using surrogate gradients (Neftci et al., 2019; Lee et al., 2020; Zhu et al., 2022) or synaptic plasticity (Bicknell & Häusser, 2021; Liu et al., 2022) requires training from scratch on large datasets, incurring high complexity, and still struggle to achieve high performance. Instead, in practice, limited training data and resources create a more urgent need to directly convert powerful ANNs into equivalent SNNs in a training-free fashion (Diehl et al., 2015). Such ANN-to-SNN conversion replaces ANN activations with temporal spike sequences, nearly preserving all capabilities of the source model. Thus, it can directly reduce the inference power consumption of open-source ANN models without other modification, even for those pretrained on large private datasets.

Nevertheless, such training-free conversion seem to be impossible for mainstream large-scale ANNs based on **Transformers** (Vaswani et al., 2017; Dosovitskiy et al., 2020; Radford et al., 2021). Their computational characteristics differs from convolutional networks, leading to two critical conflicts (Li et al., 2022). First, the matrix products between variable features in self-attention are non-causal during inference, relying on complete input spike sequences. Such multiplications are incompatible with the additive accumulation over time in SNN and thus cannot be directly calculated. Second,

---

[*]Equal contribution.    [†] Corresponding author.

unlike ReLU and BatchNorm in CNNs, operations such as GELU and LayerNorm in Transformers depend on complicated non-linearities at test-time, so that cannot be accurately represented by the quantized piece-wise linearity of spiking neurons.

Due to such inherent discrepancies, existing spiking networks cannot strictly implement Transformer operations through a directly corresponding structure. Fortunately, the spatial population coding and temporal memory properties of SNNs can be further leveraged to enhance the representational capacity on both dimensions. By redefining spiking computations as a gradual approximation process to ANN floating-point values, we propose our conversion pipeline, termed Spatio-Temporal Approximation (STA), consisting of two novel spiking modules as universal **approximators**. Spatially, we adopt the strategy of trading space for precision, introducing local neuron populations to simulate precise non-linearities through multiple discrete binary spikes. These modules are driven by synthetic data regardless of their actual input at inference for universality. Temporally, to obtain stationary spike emissions for rate-coding, we remodel the non-causal multiplications into an estimation-correction process. Based on the accumulated input memory, we first approximately estimate future reactions, then correct the results with the actual input as time progresses.

With our STA pipeline, we convert a ViT-B/32 model pretrained on CLIP (Radford et al., 2021) into an SNN. The resulting SNN directly inherits the capabilities like zero-shot classification and transferability from the large multimodal Transformer. It also achieves state-of-the-art accuracy for SNNs on multiple benchmarks after supervised fine-tuning. Additionally, our converted SNN requires no floating-point operations, enabling energy-efficient deployment on neuromorphic hardware.

In summary, our main contributions are as follows:

- We propose Spatio-Temporal Approximation (STA), a training-free pipeline to convert ANN Transformers to SNNs via universal approximations in both spatial and temporal domains.
- We provide theoretical analysis on the error bounds and convergence rates of both key modules in STA, proving their efficacy in approximating ANN computation.
- To our knowledge, we are the first to directly convert a pretrained mainstream Transformer (ViT-B/32 from CLIP) into an SNN without additional training or fine-tuning, while still retaining the generalization performance of the original model.

## 2 RELATED WORK

### 2.1 ANN-TO-SNN CONVERSION

Converting ANNs to SNNs is an active area of research for improving performance and training efficiency on large-scale tasks (Diehl et al., 2015), whereby ReLU activations in ANN are replaced by "soft-reset" IF neurons (Rueckauer et al., 2017; Han et al., 2020). Its key directions include:

**Training-free conversion** is directly conducted on pretrained ANNs through threshold balancing (Diehl et al., 2015; Rueckauer et al., 2017), parameter calibration (Li et al., 2021) and functional spike emission (Wang et al., 2022a; Li & Zeng, 2022) to convert to SNN and calibrate by only a few examples without retraining or fine-tuning. Thus, they can be applied on high-performing open-source ANN models. However, these methods are mostly limited to CNNs, lacking applicability to Transformers (Li et al., 2022) and suffering from long simulation steps.

**Training-dependent conversion** tailors the ANN for SNN compatibility before conversion (Bu et al., 2021; Ding et al., 2021; Bu et al., 2022; Jiang et al., 2023; Hao et al., 2023), or fine-tunes the SNN after conversion (Wang et al., 2022b). Despite reducing conversion loss and latency, they entail greater training costs and weaker generalization, while maintaining CNN-like structural constraints.

Our work presents a training-free approach that extends conversion beyond CNNs to Transformers. As spiking equivalents of Attention Blocks, our proposed modules approximates them spatially and temporally, thus retaining the applicability of large-scale pretrained models to complex scenarios.

### 2.2 TRANSFORMER AND SPIKE-BASED TRANSFORMER

**Transformers** have achieved impressive results on numerous tasks like natural language processing (Brown et al., 2020; Devlin et al., 2018) and computer vision (Dosovitskiy et al., 2020) via the self-attention mechanism that captures global dependencies by aggregating features across spatial

Figure 1: The modules and operators in each Residual Attention Block of ViT.

dimensions. Transformers differ from CNNs in two key aspects: **1)** interactions between spatial features, and **2)** complex non-linearity/normalization, both not achievable by existing SNNs.

**Spike-Based Transformers** are recently proposed models for direct SNN training. Li et al. (2022) substitutes the activations with spiking neurons but retains many floating-point operations. Zhou et al. (2022) intruduces a purely spiking self-attention module by modifying the Softmax operation. Zhou et al. (2023) presents the first fully event-driven Transformer through tailored residual connections. Additionally, Zhang et al. (2022a;b) design specified Transformers for event-based cameras, which do not readily extend to conventional visual data. All these models differ from ANN Transformers structurally and require training from scratch, while our method directly leverages conversion to inherit capabilities from pretrained ANN Transformers without training.

## 3 PRELIMINARIES AND PROBLEM ANALYSIS

### 3.1 NEURONS FOR ANN & SNN

In ANNs using ReLU activation, for neurons in layer $l$, we denote their output as vector $\boldsymbol{x}^l$, and the weight matrix between layer $l-1$ and $l$ as $W^l$. Ignoring bias, its floating-point inference process is:

$$\boldsymbol{x}^l = max\left(\boldsymbol{W}^l \boldsymbol{x}^{l-1}, 0\right), \quad l = 1, 2, ...T. \tag{1}$$

As for SNNs, similar to Han et al. (2020), we consider the soft-reset Integrate-and-Fire (IF) neurons. When the $l$-th layer receives weighted binary spikes $\boldsymbol{x}_s^{l-1}(t) \in \{0, 1\}$, the update rule is:

$$\boldsymbol{m}^l(t) = \boldsymbol{p}^l(t-1) + \boldsymbol{W}^l \boldsymbol{v}_{th}^{l-1} \otimes \boldsymbol{x}_s^{l-1}(t), \quad \begin{cases} \boldsymbol{s}^l(t) = H(\boldsymbol{m}^l(t) - \boldsymbol{v}_{th}^l) \\ \\ \boldsymbol{p}^l(t) = \boldsymbol{m}^l(t) - \boldsymbol{v}_{th}^l \otimes \boldsymbol{x}_s^l(t) \end{cases}, \tag{2}$$

where $\boldsymbol{m}^l(t)$ and $\boldsymbol{p}^l(t)$ represent the potentials before and after the trigger of spike $\boldsymbol{s}^l(t)$, $\boldsymbol{v}_{th}^l$ is the threshold, and $H(\cdot)$ is Heaviside step function. The firing rate is measured as the average number of spikes over time $T$, denoted as $\bar{\boldsymbol{s}}^l$. The converted SNN exhibits similarities with ReLU ANN on the activation values for each layer, i.e., $\boldsymbol{x}^l \approx \bar{\boldsymbol{s}}^l$, due to their comparable linear growth arithmetic.

### 3.2 OPERATIONS IN TRANSFORMERS

A basic attention block in Transformer is shown in Fig. 1, relying on two main types of operations that differ from those in conventional CNNs for conversion. More details on the modules in Transformer are provided in the Appendix.A.

1) **Non-linear operators.** While CNNs primarily use ReLU activation for non-linearity, Transformer involves more complex nonlinear functions like GELU (Hendrycks & Gimpel, 2016), square root, exponentiation, etc., which cannot be directly achieved by the piece-wise linear dynamics of IF neurons. This requires us to approximate their computational characteristics in the spatial domain.

2) **Variable Scalar / Matmul product.** The inference in CNNs is conducted through variable features multiplied by constant weight matrices, while Transformers contain more *variable-variable* multiplications, such as the query-key products in self-attention. Additionally, LayerNorm in Transformer computes normalization coefficients dynamically during inference, preventing integration into weight matrices as with BatchNorm in CNNs (Rueckauer et al., 2017). Thus, computing these multiplications with spiking neurons is challenging and may require temporal modifications.

## 4 SPATIAL APPROXIMATION FOR NON-LINEARITY

As Transformer's floating-point non-linearity poses challenges for SNN conversion, our goal is developing spiking counterparts to simulate their spatial reactions. The proposed approximators

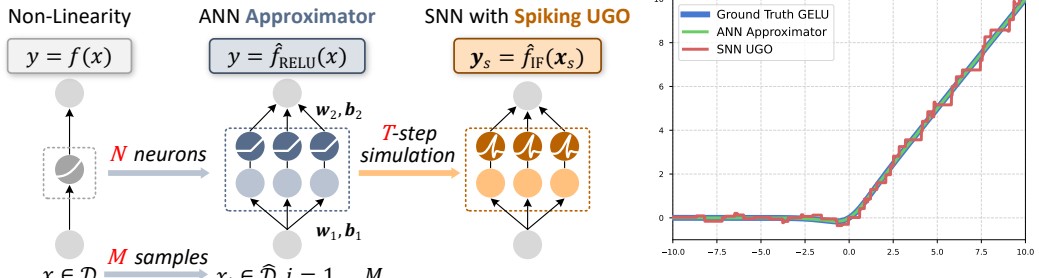

Figure 2: Spatial approximation process with UGO.

Figure 3: An approximated UGO for GELU with $N = 16, T = 16$.

should: **1)** consist only IF neurons, and **2)** be universally applicable to all operations, models and data. Due to the insufficient representation capability of each single neuron, we adopt groups of neurons to substitute individual operators. These approximators are pre-trained by synthetic floating-point data independent of real examples, and thus universally applicable to all scenarios.

### 4.1 NEURON GROUPS FOR UNIVERSAL APPROXIMATION

We first examine common non-linear operators like GELU or square root that are low-dimensional with complicated computations. We note that with the Universal Approximation Theorem (Hornik et al., 1989), single-layer ANNs can approximate these continuous functions over definite intervals. Further, ANNs with ReLU activation can be efficiently converted to equivalent SNNs. Therefore, we propose the Universal Group Operator (UGO), a small groups of spiking neurons for approximation.

**Definition 1** (Universal Group Operator). *Let $f : x \mapsto y$ defined on domain $x \in \mathcal{D}$ be a real continuous unary function. Its spiking universal group operator $\hat{f}$ comprises two fully connected (FC) layers surrounding a single hidden IF layer with $N$ neurons, such that $\exists \epsilon > 0$ where for any spike input $\boldsymbol{x}_s$ with mean $\bar{\boldsymbol{x}}_s = x$, the output spikes $\boldsymbol{y}_s$ satisfy $\mathbb{E} |\bar{\boldsymbol{y}}_s - y| \leq \epsilon$.*

*The input and output layers have weights $\boldsymbol{w}_1, \boldsymbol{w}_2 \in \mathbb{R}^n$, and biases $\boldsymbol{b}_1 \in \mathbb{R}^n, b_2 \in \mathbb{R}$, respectively.*

**Construction.** Three stages are required to obtain a universal group operator, shown in Fig. 2:

1. *Data Synthesis.* Due to LayerNorm in Transformers, the input range of any function $f$ is always empirically restricted to a small continuous interval $\mathcal{D}$, e.g., statistically, $\mathcal{D} = [-10, 10]$ for GELU. To enable the UGO to approximate $f$ without real training data, we roughly synthesize a mixture of uniform/normal distribution $\hat{\mathcal{D}}$ that covers $\mathcal{D}$, and sample $M$ points $\{x_i\}$ from $\hat{\mathcal{D}}$ to cover all possible inputs. The floating-point data pairs $\{x_i, f(x_i)\}$ serve as the synthetic training data.

2. *ANN Construction.* We manually select a suitable hyperparameter size $N$ to define the scale of an ANN $\hat{f}_n$ based on the complexity of $f$, with typically $N \in [8, 32]$ for balanced accuracy and efficiency. It is then trained on the synthetic data using ReLU or other tailored activation as in Jiang et al. (2023) for approximation.

3. *SNN Conversion.* The pretrained ANN is finally converted to an SNN $\hat{f}_{\text{IF}}$ of IF neurons over $T$ time-steps using existing methods like Li et al. (2021). Its conducts purely event-driven inference via spike accumulation and can directly replace its ANN counterpart with equivalent functionality.

The universal group operators thus allow implementation of all low-dimensional operations in Transformers for SNN conversion. As the synthesized data covers all possible inputs during inference, the pretrained UGOs are universally applicable to all test samples at high accuracy. Fig. 3 demonstrates a conversion result for GELU with $N = 16, T = 16$, and more details are in the Appendix.B.

**Approximation Error Analysis.** While bringing high efficiency, the small scale of UGOs also raise concerns about their accuracy and generalizability. To qualitatively analyze how the design impacts performance, we consider errors from three sources: insufficient sampling, limited parameterization and spiking quantization. This yields the following error bound:

**Theorem 1** (Error Bound for Spatial Approximation). *For an optimal $\hat{f}^*$, the error $\epsilon^*$ satisfies*

$$\epsilon^* \leq \underbrace{\mathcal{O}\left(\sqrt{\frac{N \log N \log M}{M}}\right)}_{Empirical\ Gap} + \underbrace{\mathcal{O}\left(\frac{\mathcal{L}_f |y|_{\max}}{N^2}\right)}_{Parameterization\ Gap} + \underbrace{\frac{\|\boldsymbol{w}_1 |x|_{\max} + \boldsymbol{b}_1\|_\infty \cdot \|\boldsymbol{w}_2\|_1}{T}}_{Quantization\ Gap}, \quad (3)$$

*where $\mathcal{L}_f$ is the Lipschitz constant of $f$ on $\mathcal{D}$. Proof in Appendix. C.*

The terms correspond to the gap between function $f$, the optimal learner, the optimal fixed-scaled ANN, and its SNN counterpart. This theoretical analysis guides our implementation in two aspects:

1. *ANN training:* The Quantization Gap reflects that the two weighted layers contribute differently to the error depending on distinct norms $\|\boldsymbol{w}_1 |x|_{\max} + \boldsymbol{b}_1\|_{\infty}$ and $\|\boldsymbol{w}_2\|_1$. Thus, unlike common $L1/L2$ regularizations, it is adopted as a layer-specific regularization during training.

2. *Hyperparameter determination:* While larger $M$ and $T$ always improve performance, the optimal scale $N$ depends on the case. Note that $\|\boldsymbol{w}_2\|_1$ can be scaled up to $N \cdot \boldsymbol{w}_{2\max}$, all three gaps correlate differently with $N$, requiring experimental search for a balance on accuracy and conversion loss.

## 4.2 Integration for High-Dimensional Operations

By proposing the universal group operator, we have achieved event-driven unary operations. However, such scheme is infeasible for normalization functions like LayerNorm and Softmax, as their higher-dimensional input space cannot be sufficiently covered by the synthesized training data as in UGOs.

To address this issue, we achieve them by integrating three types of basic spiking operations. Take LayerNorm as an example, as in Fig.4 (and Softmax in Appendix.D). The ANN implementation is $\mathrm{LN}(x_i) = \gamma \dfrac{x_i - \mu}{\sqrt{\sigma^2 + \epsilon}} + \beta$, where $\epsilon$ is a small constant, decomposed into the following parts:

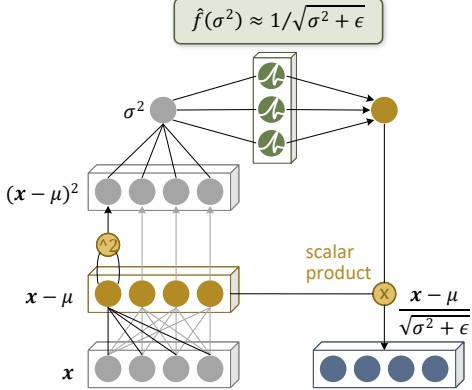

Figure 4: Integration for LayerNorm.

1. *Weighted addition:* Simple, high-dimensional computations such as zero-centering and variance for binary inputs via fixed-weight linear layers.

2. *Universal group operator:* The normalization coefficient $1/\sqrt{\sigma^2 + \epsilon}$ computed by a UGO.

3. *Multiplication:* Scalar or Matmul product between two variables, to be achieved in Section.5.

Such modular integration enables constructing high-dimensional spiking operators with UGOs, demonstrating the spatial aspect of our Spatio-Temporal Approximation pipeline. Nevertheless, performing variable multiplication in SNNs remains an unresolved issue due to its temporal characteristics. This computational requirement arises not just for normalization, but is critical for self-attention in Transformers. Therefore, we next focus on the spiking implementation of multiplications.

# 5 Temporal Approximation for Multiplications

Unlike conventional networks, the self-attention in Transformer performs multiplications between variable feature matrices rather than fixed weights. During inference, these matrices are encoded by incomplete temporal sequences, so directly computing their product is non-causal. Naively avoiding this can lead to uneven spike outputs and performance degradation. To address this, we propose Temporal-Corrective Self-Attention Layer (TCSA), employing an estimation-correction mechanism. The product is first estimated using the temporally available sequences, and then corrected by the next actual spike input. This distributes each spikes' contribution to the product across all time steps, smoothing the output for enhanced stability of multiplication.

## 5.1 Temporal Split for Spike-based Multiplication

To analysis this problem, we first consider basic matrix multiplication $\boldsymbol{A} \cdot \boldsymbol{B}$. For simplicity, assume a matrix $\boldsymbol{M}$ with shared scalar threshold $v_m$ for each element is split into a spike sequence $\boldsymbol{M}_s(t) \in \{0, 1\}, t = 1, \dots, T$. In conventional architectures, such operations typically occur between fixed-

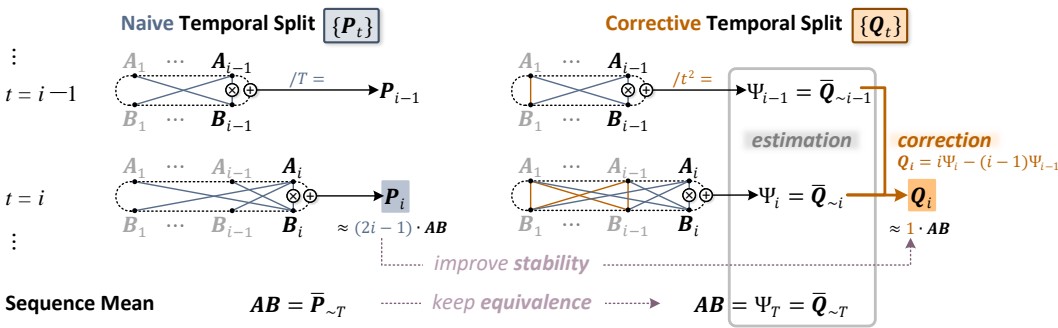

Figure 5: Spike multiplications with naive temporal split and estimation-correction mechanism.

weight matrix $\boldsymbol{W}$ and binary variable features $\boldsymbol{X}$, computed as

$$\boldsymbol{W}\boldsymbol{X} = \boldsymbol{W} \cdot v_x \bar{\boldsymbol{X}}_s = \frac{v_x}{T} \sum_{t=1}^{T} \boldsymbol{W}\boldsymbol{X}_s(t). \tag{4}$$

Thus, $v_x \boldsymbol{W}\boldsymbol{X}_s(t)$ are used as a weighted spike output at each step, and are accumulated for result.

In contrast, for common *inter-variable* multiplications in Transformer such as query-key products, the operations are rather different. Note that before the input at step $t$, both matrices are incomplete, with only inputs at $[1, t-1]$ available in their temporal split sequences.

**Definition 2** (Naive Temporal Split for Causality). *Let $\boldsymbol{A}, \boldsymbol{B}, \boldsymbol{A}_s, \boldsymbol{B}_s$ be two variable matrices and their encoded spiking sequences in $T$ steps with thresholds $v_a$, $v_b$. The temporary product $\Phi(t)$ is the sum of all currently available binary terms in the matrix product at step $t$ considering causality:*

$$\Phi(t) \triangleq \sum_{i=1}^{t} \boldsymbol{A}_s(i) \sum_{j=1}^{t} \boldsymbol{B}_s(j) = \sum_{i,j=1}^{t} \boldsymbol{A}_s(i)\boldsymbol{B}_s(j). \tag{5}$$

*Since $\Phi(t-1)$ is available before step $t$, the increment $\phi(t)$ to obtain $\Phi(t)$ is defined as below:*

$$\phi(t) \triangleq \Phi(t) - \Phi(t-1) = \boldsymbol{A}_s(t)\boldsymbol{B}_s(t) + \boldsymbol{A}_s(t)\sum_{i=1}^{t-1}\boldsymbol{B}_s(i) + \sum_{i=1}^{t-1}\boldsymbol{A}_s(i)\boldsymbol{B}_s(t), \tag{6}$$

*which uses only Boolean ANDs and additions. Accordingly, let $\boldsymbol{P}(t) \triangleq \frac{v_a v_b}{T}\phi(t)$ be the output at $t$:*

$$\bar{\boldsymbol{P}} = \frac{1}{T}\sum_{t=1}^{T}\boldsymbol{P}(t) = \frac{1}{T}\sum_{t=1}^{T}\frac{v_a v_b}{T}\phi(t) = \frac{v_a v_b}{T^2}\Phi(T) = \boldsymbol{A}\boldsymbol{B}, \tag{7}$$

*which aligns with the objective of ANN-to-SNN conversion.*

## 5.2 Estimation-Correction for Firing-Rate Stability

Although the naive method in Def.2 maintains numerical equivalence in the conversion, its output $\boldsymbol{P}(t)$ contains $2t-1$ terms due to the incomplete sequence temporarily. This implies a linearly growing magnitude over time, leading to uneven firing rates along the time dimension. As these spikes propagate, the large inputs in the last few steps make subsequent neurons hoard substantial residual membrane potential, preventing effective spike emission. To mitigate such instability, it is necessary to estimate the distribution of future input spikes earlier on, so as to react proactively.

**Methodology.** Considering the temporal consistency of rate-coding, we propose that by regarding the available sequence at $t$ as a $t$-point sampling of the complete $T$-step simulation, the overall firing rate can be approximated by that of a shorter $t$-step time interval. The estimation is thus defined as:

**Theorem 2** (Temporal Estimation). *The unbiased estimations of $\boldsymbol{A}$ and product $\boldsymbol{A}\boldsymbol{B}$ at step $t$ are*

$$\hat{\boldsymbol{A}}(t) = \frac{v_a}{t}\sum_{i=1}^{t}\boldsymbol{A}_s(i), \quad \Psi(t) = \hat{\boldsymbol{A}}(t)\hat{\boldsymbol{B}}(t) = \frac{v_a v_b}{t^2}\Phi(t), \tag{8}$$

Such estimation provides two key benefits: 1) Guaranteed evenness: As $\mathbb{E}\Psi(t) = \boldsymbol{A}\boldsymbol{B}$ for any $t$, the estimation is independent of $t$ with small temporal variation, resulting in sparse spike outputs.

2) Progressive approximation: Since $\lim_{t \to T} \Psi(t) = \Psi(T) = \boldsymbol{AB}$, the estimate gradually approximates the exact statistic for the full sequence. Each step's output brings the estimate closer to the final result. Thus, we propose:

**Definition 3** (Temporal Correction). *The corrective increment $Q(t)$ as the output sequence is:*

$$\boldsymbol{Q}(t) \triangleq t\Psi(t) - (t-1)\Psi(t-1) = \frac{v_a v_b}{t}\left[\frac{1}{1-t}\Phi(t-1) + \phi(t)\right] \tag{9}$$

*where all computations are Boolean ANDs and their weighted additions, such that*

$$\bar{\boldsymbol{Q}} = \frac{1}{T}\sum_{t=1}^{T}\boldsymbol{Q}(t) = \Psi(T) = \boldsymbol{AB}. \tag{10}$$

This mechanism is the core of our Temporal-Corrective Self-Attention Layer as a spiking self-attention module, and is also similarly adopted in Section.4.2 for multiplications. In practice, spike multiplications are always constantly weighted, e.g., $v_a \boldsymbol{A}_s(t_1)\boldsymbol{W}_A\boldsymbol{W}_B v_b\boldsymbol{B}_s(t_2)$, and the weights of additions at each step $t$ can be pre-integrated into the linear layers $\boldsymbol{W}$ before inference. Thus, the computations remain hardware friendly. Moreover, our estimation-correction algorithm allows reusing accumulated $\Phi(t)$ values from prior time steps during the update, reducing computations.

**Estimation Error Analysis.** The performance of our corrective multiplication method relies heavily on accurate estimation. We quantitatively analyzed how our estimate $\Psi$ converges to the ground truth over time steps. Considering that all multiplications are obtained from scalar multiplications, for clarity, we assume all elements are independent with a threshold $v_{th} = 1$.

**Theorem 3** (Convergence Rate of Temporal Estimation). *Assuming two independent floating-point elements $a$ & $b$, and their converted $T$-step spiking sequence follows a stationary independent process with $Ta$ & $Tb$ spikes emitted. Denote the number of arrived spikes by step $t$ as $x$, the estimated $\Psi(t)$ satisfy: (Proof in Appendix.E)*

$$\mathbb{E}\{\Psi(t)\} = ab, \qquad \mathbb{D}\{\Psi(t)\} = \frac{ab(1-a)(1-b)}{(T-1)^2}\cdot\left(\frac{T}{t}-1\right)^2 \propto \left(\frac{1}{t}-\frac{1}{T}\right)^2. \tag{11}$$

It demonstrates the estimation error decreases quadratically with $t$ initially, then stabilizes in the final few steps. This mechanism acts as a smoothing filter, providing the temporal component of our Spatio-Temporal Approximation pipeline.

## 6 IMPLEMENTATION AND EXPERIMENTS

To demonstrate the advantages of our training-free Transformer conversion approach, we apply our pipeline to the Image Encoder of CLIP (Radford et al., 2021), a prevalent Language-Image model. This allows our converted model to leverage CLIP's powerful generalization abilities such as zero-shot classification. In comparison to conventional ResNet architectures, Transformers can better exploit large-scale pretraining to achieve superior performance. Furthermore, for a fair comparison with existing methods, we fine-tune the pretrained ViT on benchmarks like CIFAR and ImageNet, achieving state-of-the-art results of SNN with smaller conversion error and faster simulation.

### 6.1 CONVERSION IMPLEMENTATION

Our work enables all Transformer computations in SNN to be conducted without specified conversion methodology. In practice, we combine prior techniques to complete the entire conversion, including MMSE (Li et al., 2021) to determine optimal neuron thresholds, signed neurons (Wang et al., 2022a) to handle negative weighted inputs, and burst spikes (Li & Zeng, 2022) to mitigate lagging inputs and reduce residual potentials. Implementation details are provided in Appendix.F.

### 6.2 ZERO-SHOT CLASSIFICATION

**Settings and Models.** CLIP is a multi-modal ANN trained on image-text pairs with diversified Image Encoder backbones including ResNet and Vision Transformer (ViT). It performs various

Table 1: Comparison with other backbones and baselines on **zero-shot** classification of CLIP.

| Dataset | Model | Method | ANN Acc. | T=32 | T=64 | T=128 | T=256 |
|---|---|---|---|---|---|---|---|
| CIFAR-10 | ResNet-50 | Calib. (Li et al., 2021) | 72.35 | 64.08 | 68.13 | 71.04 | 71.19 |
| | | SNM (Wang et al., 2022a) | | 58.69 | 61.22 | 70.68 | 70.88 |
| | ResNet-101 | Calib. (Li et al., 2021) | 79.64 | 38.21 | 55.37 | 67.44 | 71.21 |
| | | SNM (Wang et al., 2022a) | | 43.25 | 52.68 | 68.42 | 72.96 |
| | ViT-B/32 | **STA (Ours)** | 89.74 | **87.71** | **88.20** | **88.29** | **88.34** |
| CIFAR-100 | ResNet-50 | Calib. (Li et al., 2021) | 41.01 | 24.67 | 33.41 | 38.20 | 39.01 |
| | | SNM (Wang et al., 2022a) | | 35.64 | 34.71 | 39.95 | 41.13 |
| | ViT-B/32 | **STA (Ours)** | 64.26 | **62.55** | **62.74** | **62.98** | **63.01** |
| ImageNet-200 | ResNet-50 | Calib. (Li et al., 2021) | 45.63 | 22.50 | 34.51 | 41.82 | 42.03 |
| | | SNM (Wang et al., 2022a) | | 25.43 | 38.17 | 42.25 | 42.95 |
| | ViT-B/32 | **STA (Ours)** | 62.25 | **59.79** | **61.24** | **61.53** | **61.66** |
| CIFAR-10.1 | ResNet-50 | Calib. (Li et al., 2021) | 65.05 | 61.01 | 63.44 | 64.39 | 64.42 |
| | | SNM (Wang et al., 2022a) | | 44.56 | 58.26 | 63.53 | 64.06 |
| | ViT-B/32 | **STA (Ours)** | 84.15 | **83.05** | **83.25** | **83.58** | **83.52** |
| CIFAR-10.2 | ResNet-50 | Calib. (Li et al., 2021) | 63.90 | 58.97 | 61.01 | 62.50 | 62.68 |
| | | SNM (Wang et al., 2022a) | | 46.83 | 54.68 | 62.94 | 63.08 |
| | ViT-B/32 | **STA (Ours)** | 80.35 | **78.55** | **79.65** | **79.77** | **79.83** |

tasks based on natural language prompts. Since no existing methods directly convert Transformers, we use pretrained ResNet-50 backbone for our baselines. Following standard CLIP configuration for zero-shot prediction, we evaluate on CIFAR-10/100, ImageNet-200 benchmarks, and distribution-shifted CIFAR-10.1/10.2 datasets. Details in Appendix.G.1.

**Classification performance.** The results in Table.1 show that the converted ViT model substantially exceeds ResNet across all datasets and time settings. This confirms that large-scale pretrained Transformer are superior to convolutional networks for zero-shot classification, emphasizing the value of SNN conversion targeted on Transformers over CNNs.

**Accuracy loss from conversion.** Despite having more parameters than ResNet-50 (87.8M vs 25.6M), our ViT model still experiences much lower accuracy drop after conversion. Two main factors contribute: 1) Self-attention layers have lower precision requirements than convolutions, making them less prone to numerical errors. 2) Transformer architecture provides more robust features with larger label margins, maintaining predictions even under conversion perturbations.

**Limitations of existing works.** We make two key observations: 1) Larger convolutional networks like ResNet-101 do not improve SNN conversion performance over ResNet-50, as their ANN accuracy still lags behind ViT while depth exacerbates conversion errors. This highlights the need for advanced architectures like Transformers. 2) Many current conversion methods only succeed on models like resnet-20 or VGG-16, while being incompatible with deep residual networks. Thus we selectively demonstrate those with better ResNet-50 results from CLIP.

## 6.3    STANDARD CLASSIFICATION AND ABLATION STUDIES

**Standard Classification.** We fine-tune our ViT on benchmarks and compared its performance on conventional image classification tasks to resnet-20 and pretrained ResNet-50 baselines from CLIP. Table.2 shows results on CIFAR-100, with other results on CIFAR-10 / ImageNet in the Appendix.G.2. Compared to other conversion methods, our algorithm achieves near peak accuracy with fewer steps ($T = 32$ or $64$), while most baselines require over $128$ steps for optimal accuracy. The remaining small

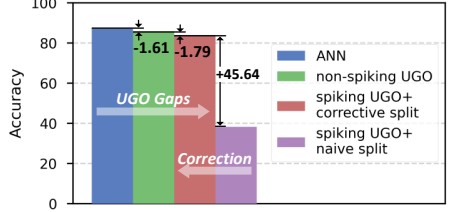

Figure 6: Ablations on components in CIFAR-100, T=32.

accuracy gap to ANN ViT is largely due to the unavoidable approximation error from the Universal Group Operators. This demonstrates the faster simulation time advantages of our approach.

**Ablations.** We also conduct ablation experiments to analyze the spatial and temporal impact in our pipeline, in Fig.6. Our results lead the the following conclusions: 1) UGO nearly eliminates the three

Table 2: Comparison with other backbones and baselines on standard classification of CIFAR-100

| Model | Method | ANN Acc. | T=32 | T=64 | T=128 | T=256 |
|---|---|---|---|---|---|---|
| resnet-20 | RMP (Han et al., 2020) | 76.12 | 30.60 | 42.61 | 62.59 | 69.86 |
| | TSC (Han & Roy, 2020) | | 35.87 | 49.70 | 65.42 | 70.59 |
| | Opt. (Deng & Gu, 2020) | | 49.81 | 69.82 | 75.75 | 75.94 |
| | Calib. (Li et al., 2021) | | 74.25 | 75.08 | 75.58 | 76.24 |
| | SNM (Wang et al., 2022a) | | 74.58 | 75.89 | 76.11 | 76.18 |
| | Burst (Li & Zeng, 2022) | | 71.14 | 75.50 | 75.89 | 76.03 |
| ResNet-50 (CLIP) | Opt. (Deng & Gu, 2020) | 81.13 | 64.48 | 71.71 | 76.67 | 79.52 |
| | Calib. (Li et al., 2021) | | 75.61 | 77.29 | 78.13 | 80.02 |
| | SNM (Wang et al., 2022a) | | 68.24 | 75.30 | 77.91 | 80.75 |
| ViT-B/32 | **STA (Ours)** | 87.35 | **84.15** | **85.25** | **85.69** | **85.98** |

Gaps in Eq.3, thereby retaining nonlinear computation capabilities after spatial approximation. 2) The estimation-correction mechanism for temporal multiplication prevents large residual potential accumulation caused by output lag, thus significantly improving performance over the naive method.

## 6.4 ENERGY ESTIMATION

The energy efficiency of SNN stems from two aspects: 1) Sparsity and event-driven computation, where only a small fraction of synapses are active during inference. 2) Low-power synaptic operations like Boolean logic and weighted additions instead of expensive floating-point operations. The consumption of ANN inference is characterized by floating-point operations ($FLOPs$) with energy cost $E_{MAC}$, while SNNs rely on synaptic operations ($SOPs$) with $E_{AC}$. Therefore, the ratio of inference energy for SNN versus ANN for a module is estimated in Rathi & Roy (2020) as:

$$\gamma = \frac{E_{SNN}}{E_{ANN}} = \frac{SOPs \cdot E_{AC}}{FLOPs \cdot E_{MAC}}, \quad with \ E_{MAC} \approx 4.6J, E_{AC} \approx 0.9J \qquad (12)$$

Using an empirical firing rate denoted as $\eta$, we analyze both components in our pipeline:

**Universal Group Operator.** A unary non-linear operator like GELU requires $FLOPs \approx 70$ primarily due to exponents in $\mathtt{tanh}$, while a UGO with $N$ neurons requires $SOPs = 2NT\eta$. For a high accuracy implementation with $N = 32, T = 32, \eta \approx 9.1\%$, UGOs reduce computational costs by $41\%$ compared to GELU. This saving is further amplified in high-dimension operations.

**Spike Multiplications.** We illustrate this with the $N \times N$ query-key matrix products, where $FLOPs = 3N^3$. While naively implementing matrix multiplication requires $O(T^2)$ spike products, our proposed TCSA layer reduces complexity to $O(T)$ with accumulated $\Phi(t)$. Specifically, $SOPs = 4TN^3\eta$. With $\eta \in [3\%, 13\%]$ at $T = 32$ across all 12 blocks, the attention modules achieve $33\%$ savings on average, up to $75\%$ for the sparsest cases.

Admittedly, due to the unique computational demands of Transformer, its energy savings from SNN conversion are not superior than convolutional spiking networks. However, our work still demonstrates potential for low power usage: training UGOs with sparsity constraints or optimizing multiplication estimations could further reduce the $\eta$ in our Spatial-Temporal Approximation pipeline. In addition, the latest hardware (Pei et al., 2019) allows utilizing both floating-point and event-driven computation synergistically, thereby further improving energy performance.

## 7 CONCLUSION AND DISCUSSION

For the first time, this paper establishes a bridge between mainstream pretrained Transformers and SNNs. By designing novel spiking operators and layers, we approximate Tranformers in both spatial and temporal dimensions in a purely event-driven fashion, breaking with convention. Since all Transformer-based models share similar computation modules, our proposed pipeline is broadly applicable to various language and vision models, including the Text Encoder in CLIP, or even Large Language Models, as our subsequent work. These pretrained large models are often transferable without additional training or fine-tuning, and our training-free conversion pipeline avoids performance degradation, promoting practical SNN usage on various downstream applications. While the converted ViT has slightly higher computations than conventional spiking CNNs, it provides stronger performance and robustness with fewer simulation steps. This enables potential energy-efficient deployment of open-source large models in the future with neuromorphic hardware.

## 8 ACKNOWLEDGMENTS

This work was supported in part by the National Key Research and Development Program of China under STI 2030——Major Projects 2021ZD0200300, and in part by the National Natural Science Foundation of China under Grant 62176133, and in part by the Tsinghua-Guoqiang research program under Grant 2019GQG0006 and in part by the Natural Science Foundation of Fujian Province, China, under Grant 2022J01656.

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

# Appendix

## A  MODULES IN TRANSFORMERS

This section provides a detailed overview of all the modules, operators, and formulas in the Residual Attention Block of the Transformer. Taking ViT-B/32 as an example, it consists of one convolutional layer, 12 sequentially connected Residual Attention Blocks, and a final linear layer. Each Attention Block contains the following modules:

1. **Multi-Head Attention** is the core module of each block, allowing the model to jointly attend to information from different representation subspaces. In self-attention, the same input vector is first projected $h$ times into queries $\boldsymbol{Q}$, keys $\boldsymbol{K}$, and values $\boldsymbol{V}$ using different learned linear projections. Attention is then performed in parallel for each projection:

$$Head_i = \text{Softmax}(\boldsymbol{Q}_i \boldsymbol{K}_i^T)\boldsymbol{V}_i, \quad for\ i = 1, ..., h \tag{13}$$

The outputs of the $h$ heads are concatenated and projected once more by $\boldsymbol{W}^O$ to get the final values:

$$MultiHead(\boldsymbol{Q}, \boldsymbol{K}, \boldsymbol{V}) = \text{Concat}(Head_1, ..., Head_h)\boldsymbol{W}^O \tag{14}$$

The uniqueness of Attention is that it performs a large number of matrix multiplications between feature matrices, such as $\boldsymbol{X} = \boldsymbol{Q}_i \boldsymbol{K}_i^T$, which is different from the multiplications with constant weight matrices like $\boldsymbol{W}^O$.

In addition, the Softmax function is applied to the result of the query-key multiplication for the attention weights in each heads. Specifically, Softmax normalizes the attention weights to output a probability distribution:

$$\text{Softmax}(x_i) = \frac{e^{x_i}}{\sum_i e^{x_i}}, \tag{15}$$

which requires nonlinear operations like exponentiation and inverse.

2. **Layer Normalization** (LayerNorm), is used twice in each block before and after the attention module to stabilize and accelerate training. It normalizes the activations of each layer by subtracting the mean and dividing by the standard deviation:

$$\text{LN}(x_i) = \gamma \frac{x_i - \mu}{\sqrt{\sigma^2 + \epsilon}} + \beta, \tag{16}$$

where $\mu$ and $\sigma^2$ are the mean and variance calculated over all hidden units in the same layer.

Notably, unlike BatchNorm which tracks global statistics during training across entire channels, LayerNorm normalizes each element independently. This normalizing occurs at both training and inference, requiring dynamic statistics, which hinders weights absorption into matrices compared to BatchNorm for SNN conversion. Therefore, it also requires more complex nonlinear operations like square root and inverse at inference.

3. **Gaussian Error Linear Unit** (GELU) is used as the non-linear activation function in the MLP in each self-attention block. It applies the cumulative distribution function of the Gaussian distribution to each input element $x_i$, and is approximated as:

$$\text{QuickGELU}(x_i) = x \cdot sigmoid(1.702x), \tag{17}$$

which allows gradients to flow efficiently through the activation during backpropagation.

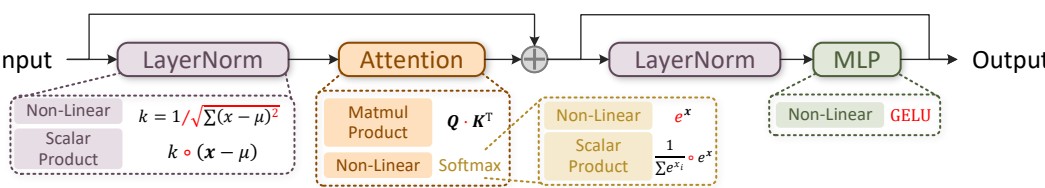

Figure 7: The modules and operators in each Residual Attention Block of ViT.

# B  SETTINGS OF UNIVERSAL GROUP OPERATOR

The Universal Group Operator achieves four main operations: 1) **Exponentiation** in Softmax, 2) **GELU** in MLP, 3) **Inverse** in Softmax, 4) **Inverse of Square Root** in LayerNorm. The settings required for training a UGO are organized as follows:

1. *Data Synthesis.* The synthesized distribution $\hat{\mathcal{D}}$ and the sample number $M$.
2. *ANN Construction.* The hidden-layer size $N$, the loss function, optimizer and scheduler.
3. *SNN Conversion.* The selected threshold $V_{th}$.

These settings are summarized in Table.3, where the Loss Penalty refers to the Quantization Gap in Eq.3. To empirically demonstrate $\mathcal{D}$ in practice, the inputs of each module under real sampling conditions are provided in Fig.8.

The fitting results of Universal Group Operators implemented in our algorighm are shown in Fig.9.

Table 3: Hyperparameters and settings for UGO training.

|  | Exp | GELU | Inverse | LayerNorm |
|---|---|---|---|---|
| $\hat{\mathcal{D}}$ | U(-35,3) 50% U(-12,2) 50% | U(-25,25) 50% U(-1,1) 50% | U(3,38) 75% U(2,75) 25% | U(0.01,1) 100% |
| $M$ | | 164384 samples * 128 batch * 1000 epoch | | |
| $N$ | 32 | 32 | 16 | 8 |
| $Loss$ | Huber | Huber+Penalty | Huber+Penalty | MSE |
| $Optim - LR$ | SGD-0.01 | SGD-0.01 | SGD-0.01 | Adam-0.01 |
| $Scheduler$ | MultiStepLR milestones=[500,800] $\gamma = 0.5$ | StepLR step=100 $\gamma = 0.5$ | MultiStepLR milestones=[500,800] $\gamma = 0.1$ | CosineAnnealingLR |
| $V_{th}$ | | Determined by Li et al. (2021) | | |

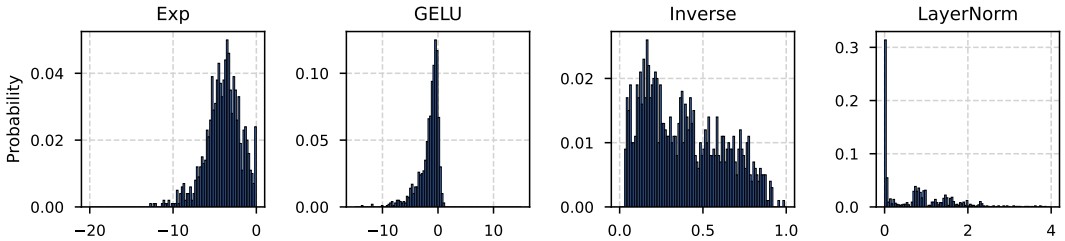

Figure 8: Empirical input distribution $\mathcal{D}$ sampled from 10 CIFAR-10 input images.

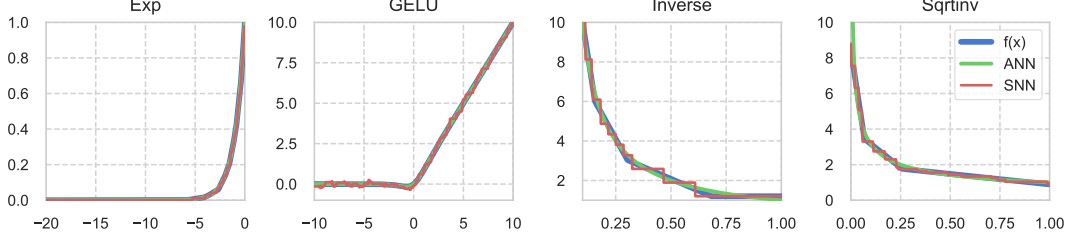

Figure 9: Fitting Results for the Universal Group Operators.

## C    PROOF FOR THEOREM 1

**Theorem 1** (Error Bound for UGO). *For an optimal $\hat{f}^*$, its approximation error $\epsilon^*$ satisfies*

$$\epsilon^* \leq \underbrace{\mathcal{O}\left(\sqrt{\frac{N \log N \log M}{M}}\right)}_{Empirical\ Gap} + \underbrace{\mathcal{O}\left(\frac{\mathcal{L}_f |y|_{\max}}{N^2}\right)}_{Parameterization\ Gap} + \underbrace{\frac{\|\boldsymbol{w}_1 |x|_{\max} + \boldsymbol{b}_1\|_\infty \cdot \|\boldsymbol{w}_2\|_1}{T}}_{Quantization\ Gap}, \quad (18)$$

*where $\mathcal{L}_f$ is the Lipschitz constant of $f$ on $\mathcal{D}$.*

*Proof.*   We decompose the error into three gaps:

- The **Empirical Gap** between $f$ and the optimal learning machine $f_m$ due to limited $M$ and model complexity regarding to $N$.

- The **Parameterization Gap** between $f_m$ and a single-layer ANN $f_n$ with $N$ neurons due to the limited parameters determined by $N$.

- The **Quantization Gap** between $f_n$ and the UGO $\hat{f}$ due to spiking discretization regarding to $T$.

**Empirical Gap.** We first quote a lemma from Bartlett et al. (2019).

**Lemma 1.** *For deep neural networks with arbitrary piecewise linear activation function where $W$ is the number of weights and $L$ is the number of layers, its VC-dimension is bounded by $\Omega(WL\log(W/L))$ and $\mathcal{O}(WL\log(W))$.*

For $f_m$, we have $L = 1$ and $W = 2N$, thus $d_{VC} = \mathcal{O}(N\log(N))$. According to the classical conclusion in Vapnik (1999), the empirical gap between $f_m$ and $f$ with $M$ samples is:

$$\epsilon_{emp} = \mathcal{O}\left(\sqrt{\frac{d_{VC} \log \frac{M}{d_{VC}}}{M}}\right) = \mathcal{O}\left(\sqrt{\frac{N \log N \log M}{M}}\right). \quad (19)$$

**Parameterization Gap.** Considering that the parameter quantity $2N$ of the ANN is much smaller than the sampling quantity $M$ in practice, the ANN $f_n$ cannot empirically fit all data, leading to a gap between $f_n$ and $f_m$. We modify the conclusion from Lu et al. (2021):

**Lemma 2.** *For deep ReLU networks with width $N$ and depth $L$ approximating $f \in C([0,1])^d$ with Lipschitz constant $\mathcal{L}_f$, the optimal approximation error is $\mathcal{O}\left(\mathcal{L}_f \cdot N^{-2/d} \cdot L^{-2/d}\right)$.*

Accordingly, the gap in our implementation for $f$ is

$$\epsilon_{parm} = |y|_{max} \cdot \mathcal{O}\left(\mathcal{L}_f \cdot N^{-2}\right) = \mathcal{O}\left(\frac{\mathcal{L}_f |y|_{\max}}{N^2}\right). \quad (20)$$

**Quantization Gap.** When converting the ANN $f_n$ to an SNN, we set the threshold as the maximum output of neurons to avoid truncation errors:

$$V_{th} = \max(\boldsymbol{w}_1 \cdot \boldsymbol{x} + \boldsymbol{b}_1) \leq \|\boldsymbol{w}_1 |x|_{\max} + \boldsymbol{b}_1\|_\infty. \quad (21)$$

The quantization error on the IF neuron outputs is $\frac{V_{th}}{T}$, so the error of the result is:

$$\epsilon_{quant} \leq \frac{V_{th} \cdot \|\boldsymbol{w}_2\|_1}{T} = \frac{\|\boldsymbol{w}_1 |x|_{\max} + \boldsymbol{b}_1\|_\infty \cdot \|\boldsymbol{w}_2\|_1}{T}. \quad (22)$$

It can also be generalized to:

$$\epsilon_{quant} \leq \frac{N |\boldsymbol{w}_2|}{T} \|\boldsymbol{w}_1 |x|_{\max} + \boldsymbol{b}_1\|_\infty \quad (23)$$

Combining Eq.19, 20, 22, the result is proved.    $\square$

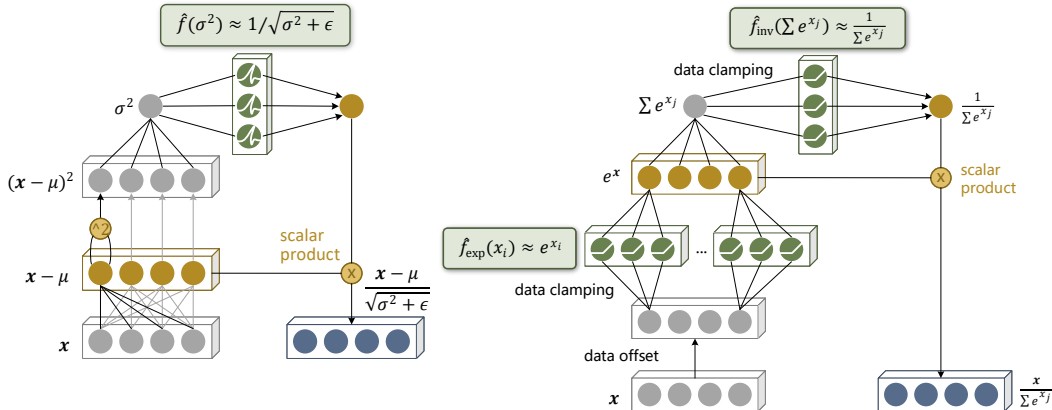

Figure 10: Integration for Layernorm.  Figure 11: Integration for Softmax.

## D  DETAILS OF HIGH-DIMENSIONAL OPERATIONS

We have roughly introduced the decomposition process of high-dimensional operations represented by LayerNorm in 4.2. In this section, the implementation details of LayerNorm and Softmax will be explained.

### D.1  LAYERNORM

In order to adapt $\mathrm{LN}(\boldsymbol{x}) = \gamma \frac{\boldsymbol{x}-\mu}{\sqrt{\sigma^2+\epsilon}} + \beta$ to SNN computations, we unroll it into the following steps:

1. Calculate the decentralized inputs $\boldsymbol{x} - \mu = \boldsymbol{x} - \frac{1}{n}\sum_{i=1}^{n} x_i = \boldsymbol{W}^{\mathrm{dc}}\boldsymbol{x}$, $x_i \in \{0,1\}$.

2. Calculate self-scalar product of $\boldsymbol{x} - \mu$, i.e. $\boldsymbol{h} = (\boldsymbol{W}^{\mathrm{dc}}\boldsymbol{x}) \circ (\boldsymbol{W}^{\mathrm{dc}}\boldsymbol{x})$, which can be implemented with TCSA in SNN.

3. Calculate the variance $\sigma^2 = \bar{\boldsymbol{h}} = \boldsymbol{W}^{\mathrm{avg}}(\boldsymbol{W}^{\mathrm{dc}}\boldsymbol{x}) \circ (\boldsymbol{W}^{\mathrm{dc}}\boldsymbol{x})$.

4. Approximate the inverse of the standard deviation via (spiking) UGO, i.e. $\frac{1}{\sqrt{\sigma^2+\epsilon}} \approx \hat{f}(\sigma^2) = v_{th}\boldsymbol{y}$, $y_i \in \{0,1\}$

5. Calculate scalar product of the inverse and the decentralized inputs with factor $\gamma, \beta$ to get $\mathrm{LN}(\boldsymbol{x}) = \gamma\frac{\boldsymbol{x}-\mu}{\sqrt{\sigma^2+\epsilon}} + \beta \approx \gamma v_{th}(\boldsymbol{W}^{\mathrm{dc}}\boldsymbol{x}) \circ \boldsymbol{y} + \beta$, implemented with TCSA in SNN.

### D.2  SOFTMAX

Like LayerNorm, Softmax can also be roughly decomposed into three suboperations: weighted summation, UGO approximation, and split multiplication. Specifically, as shown in Fig.11, Softmax is unrolled into the following steps:

1. Translate the inputs to $[-\infty, 1]$ by subtracting an offset $(v_{th}x_i)_{max} - 1$ ($v_{th}$ can be neuron-wise and $(v_{th}x_i)_{max}$ is usually in dimension 0) to ensure that no overflow occurs during the exponential operation. This translation has no effect on the result, as it will be cancelled out by the numerator and denominator in the subsequent division. Similar to LayerNorm, $x_i \in \{0,1\}$.

2. Clamp the translated inputs to a suitable range that the exponential UGO can handle, avoiding UGO output exceptions due to too small or too large input.

3. Approximate the exponential function value via (spiking) UGO, i.e. $e^{x_i} \approx \hat{f}_{\exp}(x_i) = v_{th}^{\exp}y_i^{\exp}$, $y_i^{\exp} \in \{0,1\}$.

4. Calculate $\sum_{i=1}^{n} e^{x_i} = \sum_{i=1}^{n} v_{th}^{\exp}y_i^{\exp}$.

5. Clamp $\sum_{i=1}^{n} e^{x_i}$ to a suitable range that the inverse UGO can handle.

6. Approximate the inverse via (spiking) UGO, i.e. $\frac{1}{\sum_{i=1}^{n} e^{x_i}} \approx \hat{f}_{\mathrm{inv}}\left(\sum_{i=1}^{n} e^{x_i}\right) = v_{th}^{\mathrm{inv}}y^{\mathrm{inv}}$, $y^{\mathrm{inv}} \in \{0,1\}$.

7. Calculate scalar product of the inputs and the inverse to get $\mathrm{Softmax}(\boldsymbol{x}) = \frac{\boldsymbol{x}}{\sum_{i=1}^{n} e^{x_i}} = v_{th}^{\mathrm{inv}}y^{\mathrm{inv}}\boldsymbol{x}$.

# E  PROOF FOR THEOREM 3

**Theorem 3** (Convergence Rate of Temporal Estimation). *Assuming two independent floating-point elements $a$ & $b$, their converted $T$-step spiking sequence follows a stationary independent process with $Ta$ & $Tb$ spikes emitted. Denote the number of arrived spikes by step $t$ as $x$, the estimated $\Psi(t)$ satisfy:*

$$\mathbb{E}\left\{\Psi(t)\right\} = ab, \qquad \mathbb{D}\left\{\Psi(t)\right\} = \frac{ab(1-a)(1-b)}{(T-1)^2} \cdot \left(\frac{T}{t} - 1\right)^2. \tag{24}$$

*Proof.* Considering a single scalar $a$, let $N(t)$ denote the number of spikes from sequences $a_s$ that have arrived by time $t$. Given $N(T) = Ta$, the probability of emitting $x$ spikes in the first $t$ steps is:

$$P\left(N(t) = x | N(T) = Ta\right)$$
$$= P\left(N(T) = Ta | N(t) = x\right) \cdot \frac{P\left(N(t) = x\right)}{P\left(N(T) = Ta\right)}$$
$$= P(N(T-t) = Ta - x) \cdot \frac{P\left(N(t) = x\right)}{P\left(N(T) = Ta\right)}$$
$$= \binom{T-t}{Ta-x}\binom{t}{x}\binom{T}{Ta}^{-1}. \tag{25}$$

For the expectation of $x$ and corresponding estimation $\hat{a}(t)$:

$$\mathbb{E}(x) = \sum_x x \cdot \frac{\binom{T-t}{Ta-x}\binom{t}{x}}{\binom{T}{Ta}} = t \sum_x \frac{\binom{T-t}{Ta-x}\binom{t-1}{x-1}}{\binom{T}{Ta}} = t\frac{\binom{T-1}{Ta-1}}{\binom{T}{Ta}} = t\frac{Ta}{T} = ta \tag{26}$$

$$\mathbb{E}(\hat{a}(t)) = \frac{\mathbb{E}(x)}{t} = a. \tag{27}$$

The second order is similarly derived as:

$$\mathbb{E}(x^2) = \sum_x x^2 \cdot \frac{\binom{T-t}{Ta-x}\binom{t}{x}}{\binom{T}{Ta}}$$
$$= t\left\{\sum_x (x-1)\frac{\binom{T-t}{Ta-x}\binom{t-1}{x-1}}{\binom{T}{Ta}} + \sum_x \frac{\binom{T-t}{Ta-x}\binom{t-1}{x-1}}{\binom{T}{Ta}}\right\}$$
$$= t(t-1)\frac{\binom{T-t}{Ta-x}\binom{t-2}{x-2}}{\binom{T}{Ta}} + ta$$
$$= ta(t-1)\frac{(Ta-1)}{(T-1)} + ta. \tag{28}$$

For the variance:

$$\mathbb{D}(x) = \mathbb{E}(x^2) - \mathbb{E}^2(x) = ta\frac{(1-a)(T-t)}{T-1} \tag{29}$$

$$\mathbb{D}\left(\hat{a}(t)\right) = \frac{\mathbb{D}(x)}{t^2} = \frac{a(1-a)}{T-1} \cdot \left(\frac{T}{t} - 1\right). \tag{30}$$

As the input elements $a$ & $b$ are independent in neural networks, the statistics of their product is:

$$\mathbb{E}\left(\Phi(t)\right) = \mathbb{E}\left(\hat{a}\hat{b}\right) = ab \tag{31}$$

$$\mathbb{D}\left(\Phi(t)\right) = \frac{ab(1-a)(1-b)}{(T-1)^2} \cdot \left(\frac{T}{t} - 1\right)^2. \tag{32}$$

$\square$

# F  IMPLEMENTATION OF UGO-APPROXIMATED ANN CONVERSION TO SNN

Due to the large network size of ViT, traditional methods like MaxNorm(Rueckauer et al., 2017) are not sufficient to preserve ANN's performance. Therefore we use some advanced, training-free techniques for conversion.

## F.1  THRESHOLD BALANCING

When a UGO-approximated ANN is obtained, the activation functions have all been replaced with ReLU, which is convenient for us to directly convert it to SNN. A small number of training samples are needed to obtain the maximum activation value and its quantitative estimate to determine the threshold potential of the neurons (Li et al., 2021). Our optimization objective is:

$$
\min_{\boldsymbol{v}_{th}^l} \left( \mathrm{QT}\left(\boldsymbol{z}^l, T, \boldsymbol{v}_{th}^l\right) - \mathrm{ReLU}\left(\boldsymbol{z}^l\right) \right), \qquad \mathrm{QT}\left(\boldsymbol{z}^l, T, \boldsymbol{v}_{th}^l\right) = \frac{\boldsymbol{v}_{th}^l}{T} \cdot \mathrm{clamp}\left( \left\lfloor \frac{T}{\boldsymbol{v}_{th}^l} \boldsymbol{z}^l \right\rfloor, 0, T \right),
$$
(33)

where $\boldsymbol{z}^l = \boldsymbol{W}^l \boldsymbol{x}^{l-1}$. As Eq.33 has no closed-form solution, we find the optimal threshold by grid search, enumerating $n(n = 100$ in our experiments) values within $[0.5\boldsymbol{z}_{max}^l, \boldsymbol{z}_{max}^l]$.

## F.2  SIGNED NEURONS WITH MEMORY POTENTIAL

For the IF neuronal structure that can only release positive spikes, if the input from a neuron's negative-weighted synapse arrives late and the positive-weighted input has already been converted into a spike, a portion of the significant negative potential information will not be transmitted. Therefore, we introduce Signed Neurons with Memory potentials(SNM) which allow negative spikes to be released(Wang et al., 2022a) to ensure that negative-weighted information is not lost. In this perspective, Eq.2 can be refined as:

$$
\begin{aligned}
\boldsymbol{m}^l(t) &= \boldsymbol{p}^l(t-1) + \boldsymbol{W}^l \boldsymbol{s}^l(t), \\
\tilde{\boldsymbol{r}}^l(t) &= \boldsymbol{r}^l(t-1), \\
s^{l,i}(t) &= \begin{cases} v_{th}^{l,i}, & m^{l,i}(t) \geq v_{th}^{l,i} \\ -v_{th}^{l,i}, & m^{l,i}(t) \leq -v_{th}^{l,i} \quad \text{and} \quad \tilde{r}^{l,i}(t) > 0 \\ 0, & \text{otherwise} \end{cases}, \\
\boldsymbol{p}^l(t) &= \boldsymbol{m}^l(t) - \boldsymbol{s}^l(t), \\
\boldsymbol{r}^l(t) &= \tilde{\boldsymbol{r}}^l(t-1) + \boldsymbol{s}^l(t)
\end{aligned}
$$
(34)

where $\tilde{\boldsymbol{r}}^l(t)$ and $\boldsymbol{r}^l(t)$ denotes the memory potential before and after spikes' triggering.

## F.3  BURST SPIKES WITH $\rho$-SCALE THRESHOLD

In order to minimize the effect of lagging inputs generated during SNN inference, we use the burst spikes mechanism, which allows neurons to clear off residual potentials in the form of $\Gamma(\Gamma = 2$ in our experiments) high-frequency spikes between regular emissions(Li & Zeng, 2022). The threshold of the residual potential is set to $\rho \boldsymbol{v}_{th}$. Considering its small scale in relation to $\boldsymbol{v}_{th}$ and without disrupting the quantization relationship established by Eq.33, we set $\rho = 0.5$.

## F.4  ALGORITHM

---

**Algorithm 1** STA Conversion Pipeline

---

**Input** Pretrained ANN; Non-linearities $\{f_i\}$ with synthetic distributions $\{\hat{\mathcal{D}}_i\}$; Simulation length $T$

  **for** each nonlinear function $f_i$ in Transformer **do**

    Initialize UGO model $\hat{f}_i$ with $N_i$ hidden neurons

    Sample $M$ points $\{x_j\}_{j=1,...,M}$ from $\hat{\mathcal{D}}_i$

    Optimize $\hat{f}_i$ using labels $\{f_i(x_j)\}_{j=1,...,M}$

  **end for**

  Replace non-linearites $\{f_i\}$ in pretrained ANN with $\{\hat{f}_i\}$

  Replace multiplications in pretrained ANN with TCSA (cf.Eq.9)

  **for** $l = 1, 2, ...L$-th ReLU layer in the ANN **do**

    Collect the input $\boldsymbol{x}^l$ and the output $\boldsymbol{x}^{l+1}$

    Find the optimal threshold $\boldsymbol{v}_{th}^l$ for SNN by grid search (cf.Eq.33)

  **end for**

**Output** Converted SNN

---

**Algorithm 2** STA Inference

---

**Input** Converted SNN; Simulation length $T$; Burst length $\Gamma$; Burst scale $\rho$

  **for** $t = 1, 2, ..., T$ **do**

    **for** each forward operation $g$ in the SNN **do**

      **if** $g$ is a multiplication layer **then**

        Output temporary scaled product $\sum_{j=1}^{t} k(t)\boldsymbol{X}_1(j)\boldsymbol{W}_1\boldsymbol{W}_2\boldsymbol{X}_2(j)$ using TCSA (cf.Eq.9)

      **end if**

      **if** $g$ is a non-spiking linear layer **then**

        Output $\boldsymbol{W}\boldsymbol{x}(t) + \boldsymbol{b}$

      **end if**

      **if** $g$ is a spiking linear layer constructed by IF neurons **then**

        Calculate $\triangle\boldsymbol{v} = \boldsymbol{W}\boldsymbol{x}(t) + \boldsymbol{b}$

        Release positive & negative spikes with threshold $\boldsymbol{v}_{th}$, and update potentials $\boldsymbol{v}$ (cf.Eq.34)

        **for** $i = 1, 2, ...\Gamma$ **do**

          Release spikes with threshold $\rho\boldsymbol{v}_{th}$, and update potentials $\boldsymbol{v}$

        **end for**

      **end if**

    **end for**

  **end for**

---

# G   Supplementary for Experiments

## G.1   Datasets

**CIFAR-10** is a dataset developed by the Canadian Institute for Advanced Research (CIFAR), widely used as a benchmark dataset for developing and evaluating image classification models due to its manageable size and variety of classes. It consists of 60,000 color images sampled from TinyImages Dataset(Torralba et al., 2008), divided into 50,000 training images and 10,000 testing images. There are 10 different classes in CIFAR-10, including common objects like airplanes, cars, birds, cats, etc.

**CIFAR-10.1 & CIFAR-10.2** are new testing sets for CIFAR-10, each incorporating 2,000 images from TinyImages Dataset. There are small distribution shifts between them and the original data set, which may be attributed to different generation conditions (such as illumination, angle, etc.) or adversarial attacks. Therefore, they are created to assess the robustness and generalization of models trained on CIFAR-10(Recht et al., 2018; Lu et al., 2020).

In our experiments, since there is no training set for the above two datasets, we use the training samples of CIFAR-10 to determine the threshold potentials for SNN. The high accuracy in the results demonstrates that the converted SNN not only retains the generalization ability of the pretrained model but also has the robustness to this distribution shift.

**CIFAR-100** is also a subset of TinyImages Dataset and serves as a more challenging version of CIFAR-10, consisting of 100 fine-grained classes, categorized into 20 superclasses. Like CIFAR-10, it includes 50,000 training images and 10,000 testing images.

**ImageNet** is one of the largest public image databases, containing about 14 million images labeled into 1,000 categories(the full dataset is over 20,000). Unlike TinyImages Dataset, it consists of high-resolution images from the Internet. ImageNet provides a wide range of help for the realization of tasks such as image classification, target detection and semantic segmentation in large-scale scenarios. ImageNet-200 is a well-chosen subset of ImageNet containing 200 categories that can help train and evaluate models more efficiently.

## G.2   Additional Results

We provide additional results on standard classification tasks using fine-tuned ViT-B/32 from CLIP, as well as other models like resnet-20 and ResNet-34 trained directly on these dataset. The pretrained ResNet-50 does not surpass the performance of direct training, while ViT-B/32 performs well on generalization.

To determine the optimal scale $N$ and training method for the Universal Group Operator (UGO), we conducted ablation experiments on the CIFAR-100 dataset with $T = 32$. By modifying the UGO parameters, we compared different settings' impact on overall accuracy. The results in Table.6 show that as we increased N, accuracy shows decelerated growth with efficiency continually declined. Considering both factors, we selected a balanced approach that achieves good accuracy without excessive computational cost. The ablation experiments guided our selection of an appropriate UGO scale and training method.

To verify the effectiveness of the techniques mentioned in F, we did several sets of ablation experiments on cifar-10 with $T = 32$. Table.7 shows that the SNM structure significantly improves the performance of the converted SNN. The neuron-wise search of potential threshold and the introduction of the burst spikes mechanism also play an important role in model conversion, especially when combined with SNM. This may be due to the unique dynamic characteristics caused by estimation-correction mechanism.

Table 4: Comparison with other backbones and baselines on standard classification of CIFAR-10

| Model | Method | ANN Acc. | T=32 | T=64 | T=128 | T=256 |
|---|---|---|---|---|---|---|
| resnet-20 | RMP (Han et al., 2020) | | 38.04 | 59.73 | 90.10 | 90.47 |
| | TSC (Han & Roy, 2020) | | 57.64 | 71.22 | 91.30 | 92.30 |
| | Opt. (Deng & Gu, 2020) | 95.68 | 87.30 | 92.50 | 94.32 | 95.28 |
| | Calib. (Li et al., 2021) | | 94.77 | 95.02 | 95.17 | 95.44 |
| | SNM (Wang et al., 2022a) | | 94.13 | 95.43 | 95.75 | 95.69 |
| | Burst (Li & Zeng, 2022) | | 94.92 | 95.51 | 95.40 | 95.61 |
| ResNet-50 (CLIP) | Opt. (Deng & Gu, 2020) | | 71.37 | 81.52 | 85.43 | 88.10 |
| | Calib. (Li et al., 2021) | 95.71 | 87.64 | 91.85 | 92.79 | 94.60 |
| | SNM (Wang et al., 2022a) | | 90.30 | 91.42 | 92.44 | 94.31 |
| ViT-B/32 | **STA (Ours)** | 96.16 | **95.49** | **95.74** | **95.68** | **95.82** |

Table 5: Comparison with other backbones and baselines on standard classification of ImageNet

| Model | Method | ANN Acc. | T=32 | T=64 | T=128 | T=256 |
|---|---|---|---|---|---|---|
| ResNet-34 | RMP (Han et al., 2020) | 70.64 | - | - | - | 55.65 |
| | Opt. (Deng & Gu, 2020) | 70.95 | 33.01 | 59.52 | 67.54 | 70.06 |
| | Calib. (Li et al., 2021) | 75.66 | 64.54 | 71.12 | 73.45 | 74.61 |
| | SNM (Wang et al., 2022a) | 73.30 | 55.28 | 62.72 | 65.53 | 69.31 |
| ViT-B/32 | **STA (Ours)** | 83.60 | **78.72** | **82.33** | **82.56** | **82.79** |

Table 6: Ablations for settings of Universal Group Operators classification on CIFAR-100, T=32.

| $N$ and Penalty | Exp | GELU | Inverse | LayerNorm |
|---|---|---|---|---|
| T=32, N in Table.3, Baseline Accuracy=84.15 | | | | |
| 8 | -10.46 | -5.24 | -6.43 | *+0.00* |
| 8 + Penalty | -12.62 | -4.92 | -9.76 | -6.14 |
| 16 | -4.62 | -2.43 | -0.14 | **+0.02** |
| 16 + Penalty | -4.51 | -1.41 | *+0.00* | -1.79 |
| 32 | *+0.00* | -0.40 | +0.08 | -0.24 |
| 32 + Penalty | -0.08 | **+0.00** | **+0.56** | -1.58 |
| 64 | **+0.22** | -0.03 | +0.52 | -0.05 |

Table 7: Ablations for conversion techniques on CIFAR-10, T=32

| Technique Settings | | | SNN Acc. |
|---|---|---|---|
| MMSE | Burst Spikes | Use SNM | |
| layer-wise | $\Gamma = 0$ | × | 11.35 |
| layer-wise | $\Gamma = 0$ | √ | 15.57 |
| neuron-wise | $\Gamma = 0$ | × | 13.48 |
| neuron-wise | $\Gamma = 0$ | √ | 54.32 |
| neuron-wise | $\Gamma = 2, \rho = 0.5$ | × | 19.84 |
| neuron-wise | $\Gamma = 2, \rho = 0.5$ | √ | 95.26 |

Table 8: Neuron numbers and weights in each UGO unit.

|  | GELU | Softmax-Exp | Softmax-Inv | LayerNorm |
|---|---|---|---|---|
| Non-linear Neurons of Original Units | 3072 | 768 | 1 | 768 |
| Non-linear Neurons of UGO Units | 3072+(3072×)32 | 768+(768×)32 | 16 | 768+8 |
| Weights of Original Units | 0 | 0 | 0 | 0 |
| Weights of UGO Units | 32×2 | 32×2 | 16×2 | 8×2 |

## H  PARAMETER STATISTICS OF UGO UNITS

In the ANN-SNN conversion for ResNet, the number of neurons is exactly the same as the number of RELU activation layers in the original backbone. However, in the case of ViT-B/32, replacing many "non-linear operators" with "neurons" significantly increases the required number of activations. Also, defining the number of "neurons" used in the Attention Block is challenging, because its non-linearity is based on multiplications and Softmax instead of activation functions. We provide Table.8 to clarify the required number of neurons for ViT-B/32.

It shows the number of neurons and parameters for each module and its corresponding Universal Group Operator (UGO). A ViT-B model consists of 12 blocks, each containing 1 GELU activations, 2 LayerNorm operations, and 1 Softmax operation. Since all UGOs share the same parameters, the increase in weights is minimal. However, some modules like GELU need to be applied to each input feature, resulting in a significant computational load (see numbers listed in the brackets). But since we have replaced the non-linear operation with a linear one, the actual complexity does not increase significantly. The non-linear modules originally do not involve synaptic operations, thus the number of weights for these modules is zero.

