# OpenReview forum: "Spatio-Temporal Approximation: A Training-Free SNN Conversion for Transformers"
_ICLR.cc/2024/Conference — ICLR 2024 poster_

### Official Review · Reviewer_gaB3 · 2023-10-25

**Soundness:** 3 good
**Presentation:** 4 excellent
**Contribution:** 4 excellent
**Rating:** 8
**Confidence:** 5

**Summary:**

This research presents a novel method for spiking neural networks from pretrained Transformer models (or models with multi-head attentions). The suggested Universal Group Operators and Temporal-Corrective Self-Attention Layer allow a pretrained Transformer to be converted to a completely event-driven SNN without the need for training, which holds promise for neuromorphic computing. Effectively positive experimental outcomes were obtained.

**Strengths:**

The ability to convert pretrained Transformer models into spiking neural networks without the need for training is more appealing than training Spiking Transformers directly.


For neuromorphic computing and widespread deployment, the pure implementation on spiking neurons is promising.



The universal nature of the conversion method used here makes use of linear models' capacity for global approximation. The inverse function is converted correctly.

**Weaknesses:**

In comparison to conventional artificial neural networks, there may be a slight accuracy gap due to the approximation error from Universal Group Operators.

The proposed method has only been tested on the ViT-B/32 model from CLIP; it is unknown if it can be applied to other models.

The converted models perform computations at a marginally higher rate than traditional spiking CNNs (which have more synapses and neurons).

**Questions:**

From Figure 3, I can observe severe, uneven quantization. Can you explain how this quantization affects the accuracy of the output?

How do you prepare the data for pretraining non-linear activations? For GeLU, do you record the actual responses of the ANN and train it on these activation values? For other nonlinearities (inverse, exp, layer norm), how do you pretrain?

Also, why do these nonlinearities need so many kinds of losses here (Table 3)? Can you explain why Huber loss fits exp, gelu, and inverse? And why does MSE fit the layer norm?

Besides, do you evaluate the actual increment of neurons by using UGO? I want you to give me a table reporting the difference in neuron number and weight according to the models listed in Table 1.

Are you going to release the weight of the pretrained nonlinearity? Do you test the sensitivity of changing $N$ and $T$?

I think there is a typo in equation (22). $V_{th} \Vert w_2\Vert_1$ should be $(V_{th} \Vert w_2\Vert_1)/T$.

Why do you mention setting $V_{th}$ using the strategy proposed by Li et al., who proposed to use dynamic thresholds, and demonstrating the quantization gap using the maximum activation as a threshold?

---

> ### Author Response · Authors · 2023-11-13
>
> Thank you for your thorough review and detailed feedback. We hope that our responses can address your concerns adequately.
>
> ### Weakness 1
> We acknowledge your observation. Considering the highly complex non-linear functions used in ViT, such errors may be inherently difficult to eliminate in a training-free setting.
>
> ### Weakness 2
> Our method can be directly applied to various ViT models without modification. We have conducted some experiments on ViT-B/16 and provided supplementary results on the CIFAR-10 and CIFAR-100 datasets for Table 1 (zero-shot).
>
> | Dataset   | Model      | Method              | ANN Acc. | T=32  | T=64  | T=128 | T=256 |
> | --------- | ---------- | ------------------- | -------- | ----- | ----- | ----- | ----- |
> | CIFAR-10  | ResNet-50  | Calib. (Li et al.)  | 72.35    | 64.08 | 68.13 | 71.04 | 71.19 |
> | CIFAR-10  | ResNet-50  | SNM (Wang et al.)   | 72.35    | 58.69 | 61.22 | 70.68 | 70.88 |
> | CIFAR-10  | ResNet-50  | *Combined Conversion* | *72.35*    | *63.22* | *67.05* | *71.15* | *70.92* |
> | CIFAR-10  | ResNet-101 | Calib. (Li et al.)  | 79.64    | 38.21 | 55.37 | 67.44 | 71.21 |
> | CIFAR-10  | ResNet-101 | SNM (Wang et al.)   | 79.64    | 43.25 | 52.68 | 68.42 | 72.96 |
> | CIFAR-10  | ResNet-101  | *Combined Conversion* | *79.64*    | *55.03* | *64.62* | *68.37* | *73.64* |
> | CIFAR-10  | **ViT-B/32**   | **STA (Ours)**          | **89.74**    | **87.71** | **88.20** | **88.29** | **88.34** |
> | CIFAR-10  | **ViT-B/16**   | **STA (Ours)**          | **90.83**    | **86.79** | **87.33** | **87.44** | **87.47** |
> | CIFAR-100 | ResNet-50  | Calib. (Li et al.)  | 41.01    | 24.67 | 33.41 | 38.20 | 39.01 |
> | CIFAR-100 | ResNet-50  | SNM (Wang et al.)   | 41.01    | 35.64 | 34.71 | 39.95 | 41.13 |
> | CIFAR-100 | ResNet-50 | *Combined Conversion* | *41.01* | *37.41* | *39.26* | *40.31* | *40.78* |
> | CIFAR-100 | **ViT-B/32**   | **STA (Ours)** | **64.26**    | **62.55** | **62.74** | **62.98** | **63.01** |
> | CIFAR-100  | **ViT-B/16**   | **STA (Ours)**          | **67.20**    | **63.12** | **63.45** | **63.77** | **63.75** |
>
> Our algorithm can also be applied to other ViT models of different sizes, with more experiments ongoing.
>
>
> ### Weakness 3
> Due to the incompatibility between the underlying computational logic of ANN Transformers and SNNs, the cost on spiking rate is indeed challenging to completely eliminate. We are actively working on mitigating this issue and exploring potential solutions.
>
> ### Question 1
> We appreciate your concern. In fact, our quantification is not conducted on the shown non-linear function, but the hidden layer neurons that fit the non-linear function. We will provide our answer from three perspectives:
>
> 1.	Severe Quantization: IF neurons exhibit a "stepped" input-output characteristic, where each neuron produces $T$ steps. In our fitting method, we employ a weighted sum of the outputs of $N$ neurons, resulting in a total of no more than $N*T$ quantization steps. However, due to sparsity, this value is typically much smaller, leading to severe quantization.
>
> 2.	Unevenness: We construct the training data by generating more samples within a smaller numerical range (e.g., [-1, 1]). As a result, the quantization is denser in that range compared to other ranges like [5, 10]. However, empirical observations show that this unevenness does not significantly impact the results.
>
> 3.	Accuracy: The combination weights of the Universal Group Operator are obtained through training and have a certain degree of randomness. The fitting errors observed during training do not directly reflect the practical application effects on ViT. In practice, we parallel train dozens of Universal Group Operators and evaluate them on an ultra-small test set consisting of 50 samples to select the model with the highest accuracy.
>
> ### Question 2
> The details of non-linear pretraining can be found in Appendix B. Before synthesizing the training data, we put forward 10 randomly sampled images from CIFAR-10 through the network and record the actual input and response of each module, as shown in Figure 8. Due to normalization, the activation values of almost any image passing through the network follow similar statistical patterns. However, due to the limited number of samples, training the Universal Group Operator (UGO) directly with these values would lead to overfitting. Therefore, we roughly determine the range of these activation values and manually construct a uniform distribution to cover the real distribution. This simple method has been found to meet the fitting accuracy requirements. For all non-linear operators, the training data synthesis and pretraining follow the same procedure.

---

> > ### Author Response · Authors · 2023-11-13
> >
> > ### Question 3
> > The choice between Huber loss and MSE loss is primarily driven by different accuracy requirements on "points with large numerical errors." Huber loss combines MSE and MAE, with MSE having better convergence but being more sensitive to larger values, while MAE is less sensitive. In Figure 9, we observe that the non-linear functions we want to fit often involve significant output amplitude variations, and our focus is on the fitting accuracy at smaller values. Therefore, Huber loss is usually more suitable. As for LayerNorm, its accuracy requirements are not as high, so MSE is sufficient.
> >
> > ### Question 4
> > In the ANN-SNN conversion for ResNet, the number of neurons is exactly the same as the number of RELU activation layers in the original backbone. However, in the case of ViT-B/32, replacing many "non-linear operators" with "neurons" significantly increases the required number of activations. Also, defining the number of "neurons" used in the Attention Block is challenging, because its non-linearity is based on multiplications and Softmax instead of activation functions. We provide a reference table to clarify the required number of neurons for ViT-B/32.
> >
> > |    | GELU      | Softmax - Exp                                   | Softmax - Inv                          | LayerNorm    |
> > | :-------- | ---------- | :------------------ | -------- | -------- |
> > | Number of Original Units (non-linear) | $$3072$$                 | $$768$$                 | $$1$$ | $$768$$ |
> > | Number of UGO Units (linear) | $$3072+(3072\times) 32$$ | $$768+(768\times)32$$ | $$1\times16$$ | $$768+8$$ |
> > | Number of Original Weights | $$0$$ | $$0$$ | $$0$$ | $$0$$ |
> > | Number of UGO Weights | $$32\times2$$ | $$32\times2$$ | $$16\times2$$ | $$8\times2$$ |
> >
> > The table above shows the number of neurons and parameters for each module and its corresponding Universal Group Operator (UGO). A ViT-B model consists of 12 blocks, each containing 1 GELU activations, 2 LayerNorm operations, and 1 Softmax operation.
> > Since all UGOs share the same parameters, the increase in weights is minimal. However, some modules like GELU need to be applied to each input feature, resulting in a significant computational load (see numbers listed in the brackets). But since we have replaced the non-linear operation with a linear one, the actual complexity does not increase significantly.
> > The non-linear modules originally do not involve synaptic operations, thus the number of weights for these modules is zero.
> >
> > ### Question 5
> > The weights of the non-linear modules we currently use are included in the submitted code, and we plan to release a more refined version in a future public code repository. In Appendix Table 6, we present the impact of different numbers ($N$) of neurons used for fitting on the results of our algorithm when $T = 32$. This qualitative results also apply to larger time steps, such as $T = 64$ or higher. Larger $N$ and $T$ can reduce the conversion error at the cost of more energy consumption. The algorithm presented in the main paper strikes a balance between optimal accuracy and complexity. If further experiments are desired, we can conduct them accordingly.
> >
> > ### Question 6
> > Thank you for pointing out the error. We will correct it in the revised version of the paper.
> >
> > ### Question 7
> > We appreciate your thorough review. Setting the threshold as the maximum activation in the theoretical derivation is indeed not rigorous. According to Li et al.'s dynamic threshold, $V_{th}$ is typically slightly smaller than the maximum activation value, so the upper bound we derive in Equation 21-23 still holds but becomes looser. We will update the paper to reflect this correction.
> >
> > We appreciate your review on our paper, and we will refine our paper to address the concerns raised.

---

> > > ### Comment · Reviewer_gaB3 · 2023-11-21
> > > **Response to Authors**
> > >
> > > I have read the author response and the comments of other reviews. It is claimed that this work provides a solid framework for converting Transformers into their spiking versions. The authors have clearly explained the details I am concerning. I suggest in-depth exploration for the loss used for fitting the nonlinear to further decrease the delay. Overall, this is a promising conversion work. I would increase my score as my final decision. Thank you.

---

> > > > ### Author Response · Authors · 2023-11-22
> > > >
> > > > Thank you again for your appreciation and support of our work. We are now remaining committed to further exploring improved conversion and fitting methods to deploy high-performance spiking transformers with significantly reduced time steps for larger scale practical applications. We hope this paper can serve as an interesting starting point in this field.

---

### Official Review · Reviewer_wK6t · 2023-10-29

**Soundness:** 4 excellent
**Presentation:** 3 good
**Contribution:** 3 good
**Rating:** 8
**Confidence:** 4

**Summary:**

Summary:
The paper proposes a training-free method to convert transformer to SNN platforms. It proposes universal group operators to approximate nonlinear activations and temporal-corrective self-attention layer to approximate spike multiplications. Compared to prior work, it is the first to support pretrained transformers with SNNs without training.

**Strengths:**

Strength:

1.	The idea is novel. Use multiple spiking neurons to estimate the nonlinearity functions. The temporal-corrective self-attention can achieve unbiased multiplication between two variable matrices.

2.	The convergence and error bound have solid theoretical guarantees and experimental validation.

3.	Compared to prior work with training/calibration, this work can quickly convert ViT to SNN hardware with high fidelity with a small timesteps T.

4.	It also shows the efficiency benefits compared to ANNs, which justifies the advantages of SNN-based transformer.

**Weaknesses:**

Weakness:

1.	The latency/runtime benefit of SNN-based transformer with different timesteps T needs to be compared to ANN accelerators.

2.	The proposed method can have a high fidelity with a small timestep, but there is still 1% gap compared to ANNs, even with a large T.
Compared to the training/calibration-based method, the training-free one shows a higher accuracy gap. (Table2, SNM, Calib on resnet20 can fully recover the accuracy). Can the authors comment on that?

3.	Is there any randomness in the spikes-based multiplications given the current data encoding? If there is, the output of the computing result is not deterministic. How robust is it to randomness? To have a deterministic output, the effective resolution will be reduced, thus harming accuracy. Can the authors comment on that?

4.	How does the spiking-based multiplication differentiate from the standard multiplication mechanism in stochastic computing?

5.	There are other acceleration methods to speed up and reduce energy consumption by a large factor without sacrificing accuracy, e.g., model compression and better architecture design. Moving to a new hardware platform with 30-40% energy reduction seems not very convincing.

**Questions:**

Questions are listed in the weaknesses part.

---

> ### Author Response · Authors · 2023-11-13
>
> Thank you for acknowledging our work and bringing up important points for discussion. We appreciate the opportunity to address these concerns and provide further clarification.
>
> ### Weakness 1:
> As this paper primarily focuses on algorithm development on neuromorphic platforms, our knowledge regarding Transformers on ANN accelerators is limited. We plan to promptly engage with our hardware collaborators to conduct a more comprehensive analysis if possible. Besides, we provide a more detailed explanation of the theoretical energy consumption calculation method used in our work in the response to Weakness 5.
>
> ### Weakness 2:
> We acknowledge the phenomenon you have pointed out. This gap primarily stems from the structural differences between the two ANN backbones, ResNet and Transformer. In ResNet, the non-linearities can be represented as piecewise linear functions, resulting in quantization and truncation errors that cause only minor accuracy loss. However, the non-linear operations in Transformer, such as GELU, softmax, and exp, are more complex. Consequently, we employ a group of neurons (UGOs) to fit and approximate these complex functions. The errors introduced during this approximation can never be recovered by increasing timesteps, but can be further optimized during pre-training UGOs.
>
> ### Weakness 3
> From an algorithmic perspective, with fixed input values, the existing encoding methods adhere to deterministic arithmetic rules. Consequently, the spike results of the input and output on each neuron are always deterministic. Our algorithm only focuses on the rate statistics at the end of all time steps, and the temporal resolution is solely used to adjust the quantization accuracy of spike encoding without other significant affect.
>
> ### Weakness 4
> We appreciate your insightful observation. Indeed, both stochastic computing and SNN multiplications share the computational objective of encoding firing rates with two bit streams and performing multiplications on them. However, in stochastic computing, the result is influenced by the correlation between the occurrence positions of spikes in the two sequences, leading to instability. In SNNs, where spikes are often sparse, this instability is more pronounced and can result in significant errors, particularly in deep networks when timesteps are insufficient. Our proposed algorithm addresses this challenge by achieving stable and accurate computation of the multiplication of the two sequences, albeit at the cost of increased computation compared with stochastic computing.
>
> ### Weakness 5
> The energy reductions mentioned (30%~40%) are not based on actual measurements on a hardware platform, but rather serve as reference values at the algorithmic level. These estimates facilitate performance comparisons with existing SNN conversion works. Specifically, we adopt the typical energy values proposed in the 2014 paper [1], such as $E_{MAC}=4.6pJ, E_{AC}=0.9pJ. These values are outdated as hardware has evolved since then. However, for fair comparisons, many mainstream works in the field, such as [2], [3], [4], still employ these values as the standard. Furthermore, the energy reductions achieved by these existing conversions also ranges from 30% to 50%.
>
> Our contribution primarily lies on enabling the direct deployment of ANN Transformers on neuromorphic hardware. In order to directly transfer pre-trained model parameters, we could not redesign the model structure to optimize energy consumption. However, we may explore some model compression methods in the future that do not compromise the structure.
>
> Thank you again for your valuable feedback, which has allowed us to further elaborate on these weaknesses and provide additional context for our research. We appreciate your time and consideration.
>
> [1] Horowitz, Mark. "1.1 computing's energy problem (and what we can do about it)." 2014 IEEE international solid-state circuits conference digest of technical papers (ISSCC). IEEE, 2014.
>
> [2] Li, Yuhang, et al. "A free lunch from ANN: Towards efficient, accurate spiking neural networks calibration." International conference on machine learning. PMLR, 2021.
>
> [3] Wang, Yuchen, et al. "Signed neuron with memory: Towards simple, accurate and high-efficient ann-snn conversion." International Joint Conference on Artificial Intelligence. 2022.
>
> [4] Li, Yang, and Yi Zeng. "Efficient and accurate conversion of spiking neural network with burst spikes." arXiv preprint arXiv:2204.13271 (2022).

---

> ### Comment · Reviewer_wK6t · 2023-12-04
> **Thanks for the response**
>
> The responses mostly addressed my questions. I raised the score to 8.

---

### Official Review · Reviewer_S5Cb · 2023-10-30

**Soundness:** 2 fair
**Presentation:** 2 fair
**Contribution:** 2 fair
**Rating:** 6
**Confidence:** 4

**Summary:**

This paper proposed Universal Group Operator (UGO) and Spatio-Temporal Approximation (STA) to fit the functions of LayerNorm, GELU layers and optimize the conversion error about self-attention modules.

**Strengths:**

1. The theoretical analysis about Temporal Estimation & Correction in Eq.5-Eq.11 is convincing.

**Weaknesses:**

1. From Tab.1-2, it seems that the author's approximate fitting methods for nonlinear operations such as LayerNorm require relatively long time-steps ($\geq 32$) to be effectively implemented, which will result in more significant time latency and energy consumption. In addition, even under 256 time-steps, the author's approximate fitting and error correction methods cannot completely eliminate the conversion error (there is a ~1% accuracy loss).

2. Regarding the fitting calculation of LayerNorm and Softmax layers, as well as the self-attention layer error correction calculation in Eq.9, it seems that the calculation steps and costs involved are still relatively large. I think this may hinder the algorithm's practical application.

3. I noticed that a previous work [1] achieved similar ANN-SNN Conversion performance to this paper when using BatchNorm layers directly (without involving nonlinear operations) and without error correction for attention modules. So I think the value of this approximate fitting and error correction method still needs to be further evaluated.

[1] Ziqing Wang, Yuetong Fang, Jiahang Cao, Qiang Zhang, Zhongrui Wang, Renjing Xu. Masked Spiking Transformer. ICCV 2023.

**Questions:**

See Weakness Section.

---

> ### Author Response · Authors · 2023-11-13
>
> Thank you for your valuable feedback on our work. Since the submission of our paper, we have made some improvements to address the issues raised.
>
> ### Weakness 1:
> 1.	Time Latency: The latency observed during ANN-SNN conversion is not solely caused by the approximation method proposed in our work. It primarily stems from the inherent requirement for higher precision in large-scale pre-trained models, which is far less sparse than typical SNN models. Additionally, we would like to highlight that in our subsequent experiments, our method also achieve satisfactory outcomes at T=16. Notably, the performance reduction of our approach is even lower that of the ResNets at the same time step.
>
> | Dataset   | Model      | Method              | ANN Acc. | T=16 | T=32  | T=64  | T=128 | T=256 |
> | --------- | ---------- | ------------------- | -------- | ----- | ----- | ----- | ----- | --------- |
> | CIFAR-10  | ResNet-50  | Calib. (Li et al.)  | 72.35    |59.71  | 64.08 | 68.13 | 71.04 | 71.19 |
> | CIFAR-10  | ResNet-50  | SNM (Wang et al.)   | 72.35    |/  | 58.69 | 61.22 | 70.68 | 70.88 |
> | *CIFAR-10* | *ResNet-50* | *Combined Conversion* | *72.35*  |/  | *63.22* | *67.05* | *71.15* | *70.92* |
> | CIFAR-10  | ResNet-101 | Calib. (Li et al.)  | 79.64    |31.65  | 38.21 | 55.37 | 67.44 | 71.21 |
> | CIFAR-10  | ResNet-101 | SNM (Wang et al.)   | 79.64    |/  | 43.25 | 52.68 | 68.42 | 72.96 |
> | *CIFAR-10* | *ResNet-101* | *Combined Conversion* | *79.64*  |/  | *55.03* | *64.62* | *68.37* | *73.64* |
> | **CIFAR-10** | **ViT-B/32** | **STA (Ours)**  | **89.74** | **83.92** | 87.71 | 88.20 | 88.29 | 88.34 |
> | CIFAR-100 | ResNet-50  | Calib. (Li et al.)  | 41.01    |22.53  | 24.67 | 33.41 | 38.20 | 39.01 |
> | CIFAR-100 | ResNet-50  | SNM (Wang et al.)   | 41.01    |/  | 35.64 | 34.71 | 39.95 | 41.13 |
> | *CIFAR-100* | *ResNet-50* | *Combined Conversion* | *41.01* |/  | *37.41* | *39.26* | *40.31* | *40.78* |
> | **CIFAR-100** | **ViT-B/32** | **STA (Ours)** | **64.26** | **54.12** | 62.55 | 62.74 | 62.98 | 63.01 |
> | ImageNet-200     | ResNet-50 | Calib. (Li et al.) | 45.63 | /          | 22.50 | 34.51 | 41.82 | 42.03 |
> | ImageNet-200     | ResNet-50 | SNM (Wang et al.) | 45.63 | /          | 25.43 | 38.17 | 42.25 | 42.95 |
> | **ImageNet-200** | **ViT-B/32** | **STA (Ours)** | **62.25** | **44.72** | 59.79 | 61.24 | 61.53 | 61.66 |
> | **ImageNet** | **ViT-B/32** | **STA (Ours)** | **57.93** | **46.62** | 55.50 | 56.39 | / | / |
>
> 2.	Conversion Error under large time-steps: The error in our method is caused by the approximation error of nonlinear operators with our Universal Group Operators (UGOs), as in Fig.6. The limited number of neurons are not sufficient to accurately fit complex functions, so such error cannot be eliminated even with infinite timesteps. Further optimization of the pre-training process of UGO can help alleviate this issue.
>
> ### Weakness 2
> Compared to convolutional networks, the ANN-SNN conversion process for Transformers does indeed incur higher computational costs. The main contribution of our work lies in demonstrating the feasibility of this conversion and exploring its computational cost in preliminary terms. In practical applications, there are various energy optimization methods available, such as spatial pruning and model compression, as well as temporal approaches based on the dynamics of different layers in the network. Furthermore, it is important to note that our SNN design is tailored specifically for hardware implementation, as we have removed nonlinear operators that hinder efficient hardware inference. Such operators are implicitly retained in many other SNN works.
>
> ### Weakness 3
> The key distinction between our work and [1] is that [1] modifies and trains the original ANN model before conversion, which is not involved in our approach. Our goal is to rapidly obtain a fully equivalent SNN counterpart for widely used large-scale pre-trained ANN models and deploy them directly in production environments. This necessitates adopting a training-free approach to prevent any degradation in generalization performance caused by fine-tuning. Furthermore, the reason for not using Batch Normalization is that it is not commonly adopted in mainstream pre-trained Transformer models.
>
> We believe that our reply addresses part of your concerns, and obtaining a higher rating is crucial for us to further optimize our research.

---

> > ### Comment · Reviewer_S5Cb · 2023-11-23
> >
> > Thanks for the response. I think the authors have effectively addressed my concerns and I have increased my rating to 6.

---

### Official Review · Reviewer_jaAV · 2023-10-31

**Soundness:** 3 good
**Presentation:** 3 good
**Contribution:** 4 excellent
**Rating:** 6
**Confidence:** 4

**Summary:**

This study introduces a training-free method to convert ANN transformers into SNNs, preserving the weights of the original pretrained model to ensure its inference capability remains intact. The resulting SNN Transformer model outperforms its convolutional network counterparts.

**Strengths:**

1.This paper overcomes the differences in computational paradigms between ANN and SNN Transformers, and can accurately approximate ANNs with converted models.

2.The proposed training-free conversion strategy could enable the direct deployment of large-scale pretrained ANN models to low-power neuromorphic hardware.

**Weaknesses:**

1.The proposed Universal Group Operators use extensive spiking neurons to model fine-grained ANN operations. This high model complexity reduces power efficiency and incurs large memory usage.

2.The current implementation only handles image Transformers. Longer sequences in language models may introduce more unaddressed issues such as threshold variations similar to that in spiking RNN.

**Questions:**

1. Would employing model compression strategies, such as pruning, enhance the efficiency and reduce the size of the Universal Group Operators?

2. The conversion implementation integrates multiple existing ANN-SNN conversion algorithms, including SNM and Burst. Using the same combined conversion for ResNet baselines could enable a more fair comparison.

3. What hurdles might one encounter when adapting this technique to language Transformers? Would the method necessitate modifications?

4. What challenges might arise when adapting this method to larger-scale transformer models?

---

> ### Author Response · Authors · 2023-11-13
>
> Thank you for your valuable feedback and acknowledgment of our work. We appreciate the opportunity to address your concerns and provide further clarification on the points raised.
>
> ### Weakness 1
> We acknowledge your observation on computational cost. However, considering that the operations themselves in ViT are more complex compared to basic ReLU neurons, a consequent decrease in efficiency may be inevitable. Besides, we actively utilize larger memory to achieve higher accuracy within limited timesteps. Further optimization can be conducted to control the power efficiency and memory cost through sparsification on the converted model.
>
> ### Weakness 2 & Question 3
> The method we proposed is also applicable to language models since the modules used in both vision/language Transformer architecture are identical. Furthermore, our approach avoids the issues caused by the sequential order of tokens in RNNs, which are absent in the Transformer. Nevertheless, we would like to highlight two practical considerations:
>
> 1.	Input sample size: In ViT-B/32, the number of patches is 7x7+1=50, whereas mainstream language models typically require 512 or more tokens. This disparity results in an increase in the scale of matrix multiplication and memory requirements.
>
> 2.	Latency of generative models: The precision of Spike Neural Network (SNN) outputs depends on the number of time steps. Consequently, generative models like GPT heavily rely on a high number of time steps to achieve accurate language generation.
>
> ### Question 1
> You are correct in suggesting that Universal Group Operators (UGO) can be further compressed. In fact, in the UGO model utilized in our paper, a significant portion of the IF neurons were rarely activated. This perspective indicates that the non-linear operations in ViT can be compressed and represented by smaller models. In future work, we plan to optimize these operations targeting language Transformers.
>
> ### Question 2
> We conducted these evaluations on the CIFAR-10 and CIFAR-100 datasets, supplementing the results in Table 1. The additional results are as follows:
>
> | Dataset   | Model      | Method              | ANN Acc. | T=32  | T=64  | T=128 | T=256 |
> | --------- | ---------- | ------------------- | -------- | ----- | ----- | ----- | ----- |
> | CIFAR-10  | ResNet-50  | Calib. (Li et al.)  | 72.35    | 64.08 | 68.13 | 71.04 | 71.19 |
> | CIFAR-10  | ResNet-50  | SNM (Wang et al.)   | 72.35    | 58.69 | 61.22 | 70.68 | 70.88 |
> | CIFAR-10  | ResNet-50  | *Combined Conversion* | *72.35*    | *63.22* | *67.05* | *71.15* | *70.92* |
> | CIFAR-10  | ResNet-101 | Calib. (Li et al.)  | 79.64    | 38.21 | 55.37 | 67.44 | 71.21 |
> | CIFAR-10  | ResNet-101 | SNM (Wang et al.)   | 79.64    | 43.25 | 52.68 | 68.42 | 72.96 |
> | CIFAR-10  | ResNet-101  | *Combined Conversion* | *79.64*    | *55.03* | *64.62* | *68.37* | *73.64* |
> | CIFAR-10  | ViT-B/32   | **STA (Ours)**          | **89.74**    | **87.71** | **88.20** | **88.29** | **88.34** |
> | CIFAR-100 | ResNet-50  | Calib. (Li et al.)  | 41.01    | 24.67 | 33.41 | 38.20 | 39.01 |
> | CIFAR-100 | ResNet-50  | SNM (Wang et al.)   | 41.01    | 35.64 | 34.71 | 39.95 | 41.13 |
> | CIFAR-100 | ResNet-50 | *Combined Conversion* | *41.01* | *37.41* | *39.26* | *40.31* | *40.78* |
> | CIFAR-100 | ViT-B/32   | **STA (Ours)** | **64.26**    | **62.55** | **62.74** | **62.98** | **63.01** |
>
> In comparison to existing baselines, this conversion approach did not yield significant performance improvements on ResNets. Moreover, it is important to note that even if it achieves accurate conversion, the performance still lags far behind ViT due to the limitations of a weak backbone.
>
> ### Question 3
> Please refer to the response provided for Weakness 2, as it addresses the applicability of our method to language models.
>
> ### Question 4
> This question is also related to Weakness 2. The main engineering challenges we encountered include the increased memory requirements resulting from larger input sizes and wider blocks, as well as the error accumulation introduced by deeper network architectures. Despite these challenges, our conversion method remains applicable since the basic modules are consistent with that of ViTs.
>
> We will continue to refine our work based on your valuable feedback. Thank you for your time and consideration.

---

> > ### Comment · Reviewer_jaAV · 2023-11-22
> > **Responses**
> >
> > Thanks for the response.
> > The authors have addressed my concerns. I recommend acceptance.

---

### Meta-Review · Area_Chair_BVUH · 2023-12-10

**Metareview:**

The paper presents a novel approach to converting pretrained Transformer models into SNNs without retraining or fine-tuning. This is significant as SNNs are known for their energy efficiency. The paper introduces two key concepts: Universal Group Operators (UGO) for spatial approximation of non-linear operations and Temporal-Corrective Self-Attention Layer for temporal approximation at inference.

Strengths:
- The paper successfully tackles the challenge of converting pretrained Transformers to SNNs.
- The conversion technique holds promise for deployment on low-power neuromorphic hardware.
- The method's universal nature allows for broad applicability across various Transformer models.

Weaknesses:
- The Universal Group Operators use a large number of spiking neurons, potentially impacting power efficiency and memory usage.
- The current implementation is limited to image Transformers, and it's unclear how it would perform with larger-scale models or language Transformers.
- There remains an accuracy gap between the converted SNNs and the original network, particularly in handling complex non-linear operations like GELU and softmax.
- Additional evaluations on a wider range of models and datasets would strengthen the paper's claims.

The AC and all reviewers vote for the acceptance of the paper.

**Justification For Why Not Higher Score:**

The method proposed in the paper has to scale to larger models to justify a higher rating.

**Justification For Why Not Lower Score:**

The technique proposed is novel and might have significant future benefits!

---

### Decision · Program_Chairs · 2024-01-16

Accept (poster)